# A unified model for interpretable latent embedding of multi-sample, multi-condition single-cell data

Ariel Madrigal [1,2], Tianyuan Lu [3,4,6], Larisa M. Soto [1,2] & Hamed S. Najafabadi [1,2,5] ✉

Single-cell analysis across multiple samples and conditions requires quantitative modeling of the interplay between the continuum of cell states and the technical and biological sources of sample-to-sample variability. We introduce GEDI, a generative model that identifies latent space variations in multi-sample, multi-condition single-cell datasets and attributes them to sample-level covariates. GEDI enables cross-sample cell state mapping on par with state-of-the-art integration methods, cluster-free differential gene expression analysis along the continuum of cell states, and machine learning-based prediction of sample characteristics from single-cell data. GEDI can also incorporate gene-level prior knowledge to infer pathway and regulatory network activities in single cells. Finally, GEDI extends all these concepts to previously unexplored modalities that require joint consideration of dual measurements, such as the joint analysis of exon inclusion/exclusion reads to model alternative cassette exon splicing, or spliced/unspliced reads to model the mRNA stability landscapes of single cells.

Single-cell technologies have emerged as powerful methods for unraveling tissue cellular heterogeneity and studying molecular phenotypes at the resolution of individual cells. Despite their remarkable potential, extracting meaningful biological insights from single-cell readouts still poses major analytical challenges. Such challenges arise from the need to integrate multiple interrelated tasks, including data normalization[1], denoising[2], and inter-sample harmonization and/or identification of a shared low-dimensional space[3,4]. These processing steps are intertwined with the analytical goals of comparing biological conditions[5] or experimental perturbations[6], extracting pathway-level activity metrics, or studying gene regulatory networks (GRNs)[7,8].

While existing methodologies have made significant strides in addressing these challenges individually, a model that can unify all these concepts into a single framework has remained elusive. For instance, the most common workflow for differential gene expression (DGE) analysis requires sequential application of normalization, inter-sample integration, and cell-type/cluster identification, to then perform a pseudo-bulk DGE analysis for each cell type across conditions of interest[5,9,10]. In addition to being limited to the analysis of discrete cell clusters, this sequential approach ignores the interplay between inter-sample integration and DGE analysis: integration depends on gene expression shifts across conditions, and DGE identification depends on integration. This is also true for other downstream analyses, such as pathway and GRN activity estimation, that are mutually influenced by normalization, low-dimensional projection, and integration steps. For example, considering prior biological information, in the form of gene networks or functionally related gene sets, may lead to identification of interpretable latent factors in expression data and help deconvolve biological variability from technical noise[8].

[1]Department of Human Genetics, McGill University, Montreal, QC H3A 0C7, Canada. [2]Victor P. Dahdaleh Institute of Genomic Medicine, Montreal, QC H3A 0G1, Canada. [3]Lady Davis Institute for Medical Research, Montreal, QC H3T 1E2, Canada. [4]Department of Statistical Sciences, University of Toronto, Toronto, ON M5S 1A1, Canada. [5]McGill Centre for RNA Sciences, McGill University, Montreal, Canada. [6]Present address: Department of Population Health Sciences, University of Wisconsin-Madison, Madison, WI 53726, USA. ✉e-mail: hamed.najafabadi@mcgill.ca

Furthermore, existing tools are primarily designed to perform such analyses on the gene expression space or other modalities in which the biological quantity of interest, e.g., mRNA abundance, is connected to a single type of observation, e.g., unique molecular identifier (UMI) counts. An array of biological processes, however, are better measured as the ratio of two quantities. For example, alternative cassette exon splicing is commonly quantified as the ratio of abundances of isoforms that include the exon vs. isoforms in which the exon is skipped[11]. As another example, the ratio of spliced to unspliced transcript abundances is informative about the processing and/or decay rates of mRNAs[12,13]. Analysis of such ratio-based modalities are particularly challenging when the observed data from the two opposing quantities are sparse (e.g., sparse UMI counts for each of the spliced and unspliced forms of a transcript). Few methods can perform dimensionality reduction on the latent space of ratio-based modalities[14] and, to our knowledge, no method exists for their inter-sample harmonization or GRN analysis.

Here, we introduce Gene Expression Decomposition and Integration (GEDI), a framework for multisample, multi-condition single-cell analysis. GEDI incorporates various single-cell analysis steps within a unified Bayesian framework that includes data integration across samples/conditions, data imputation and denoising, cluster-free DGE analysis, as well as pathway and GRN activity analysis. GEDI is competitive with other top-performing integration tools, while uniquely capable of deconvolving the effects of multiple technical and/or biological sources of sample-to-sample variability. This ability enables a natural and efficient approach for cluster-free DGE analysis by identifying the transcriptomic vector field associated with sample-level variables. Furthermore, by incorporating information about gene sets, it identifies axes of heterogeneity that are aligned with prior biological knowledge, thus enabling single-cell projection of pathway and/or GRN activities as well as their direction of change (gradients). Finally, GEDI is the first single-cell analysis framework to expand all these concepts from the gene expression space to the analysis of ratio-based modalities, including the analysis of the latent spaces of alternative splicing and RNA stability.

## Results

### The GEDI framework

We formulate multi-sample scRNA-seq analysis as the identification of sample-specific, invertible decoder functions such that the decoder function of each sample can reconstruct the expected expression profile of each cell from a (low-dimensional) representation of its biological state (Fig. 1a). Subsequently, correspondences between cells of different samples can be established based on the similarity of their biological states, as given by the inverse of the sample-specific decoders (encoders), enabling horizontal integration of single-cell data across different samples. We further constrain the sample-specific decoders to be from the same family of functions, while allowing for sample-specific parameterizations (Fig. 1b).

The decoder parameters can be optionally expressed as a probabilistic function of sample-level variables, resulting in a distribution of decoder functions for any given combination of sample characteristics (Fig. 1c). This formulation gives rise to a hierarchical generative model, in which a probabilistic function connects the characteristics of each sample to a distribution of decoder parameter sets. The parameter set of the sample is then drawn from this distribution, leading to a decoder function that connects the biological state of each cell to its expected gene expression profiles. This hierarchical model enables cluster-free differential gene expression analysis along the continuum of cell states, as we can examine how changes in sample-level variables impact the expected (mean) expression profile of any given biological cell state (Fig. 1d).

We note that the decoder function of each sample effectively defines the manifold representing the observed cell expression profiles within that sample; therefore, this formulation holds the potential for extension

to any parametric manifold learning approach. GEDI is a specific application of this general formulation, where the gene expression manifold of each sample is modeled as a hyperplane or hyperellipsoid, defined by a common (reference) set of principal axes (Fig. 1e) and sample-specific transformations of these axes (Fig. 1f). These sample-specific transformations can, in turn, be modeled as probabilistic functions of sample-level variables (Fig. 1g), enabling cluster-free analysis of the association between gene expression and sample characteristics. Optionally, the common coordinate frame can also be expressed as a probabilistic function of gene-level variables such as gene-set memberships (Fig. 1h), aligning the principal axes of the coordinate frame with prior biological knowledge and enabling the projection of pathway and regulatory network activities onto the cellular state space (Fig. 1i).

Finally, we can connect each point on the sample-specific manifolds to different types of observations using diverse data-generating distributions. This versatility enables gene expression analysis based on normalized or raw unique molecular identifier (UMI) counts, analysis of alternative splicing using counts of reads that support opposing splicing events, and analysis of RNA stability based on the UMI counts of spliced and unspliced transcripts (Fig. 1j). The GEDI model is fitted to these observation types using an expectation-maximization algorithm (see Supplementary Methods for details).

### GEDI captures different sources of sample-to-sample variability

To assess the ability of GEDI to capture sample-to-sample variability, we applied it to a dataset of peripheral blood mononuclear cells (PBMCs) from two donors[15] profiled using different scRNA-seq technologies. We applied GEDI without including sample-level variable information, so the model was oblivious to the biological and technical characteristics of the samples. Examining the sample-specific transformations learned by GEDI revealed that, once the effect of technology is regressed out, they are more similar among samples that are from the same donor (Fig. 2a). Similarly, after regressing out the effect of donor, the sample-specific transformations cluster by the technology (Supplementary Fig. 1a). These results suggest that GEDI properly learns sample-specific parameters that capture different sources of inter-sample variability, including the biological differences between the two donors and the technical variability introduced by the use of different technologies. At the same time, intra-sample variability across the cells is preserved, as the projection of the cell state representations learned by GEDI shows clear separation of cell types without any obvious separation by sample (Fig. 2b).

To quantitatively measure the ability of GEDI to separate intra-sample and inter-sample sources of variability, we compared it against a panel of existing single-cell integration methods using previously established metrics[16,17] and three benchmarking references: the PBMC dataset described above, a pancreas dataset[18,19] and the Tabula Muris[20] dataset. Overall, we observed that GEDI was consistently among the top-performing methods, regardless of the number of the latent factors used for low-dimensional projection of data—an often arbitrary choice that affects the performance of most other methods (Fig. 2c and Supplementary Fig. 1b-f). These results suggest that the manifold transformations learned by GEDI explain most of the sample-to-sample variability while retaining the heterogeneity in the biological states of the cells.

Encouraged by the performance of GEDI on PBMC data, we applied it to a recent single-cell atlas of PBMCs that included healthy individuals as well as mild and severe COVID-19 cases from two separate cohorts[21]. Consistent with the results shown above, when modeling the gene expression manifold of the cells, GEDI learned sample-specific transformations that reflected the biological variability among samples, such as the COVID-19 status and its severity (Fig. 2d). In addition, we successfully trained support vector machine (SVM) models capable of predicting the disease status (COVID-19 vs. healthy) based on sample-specific transformations of the reference manifold. Cross-cohort validation analysis suggests that, when trained on cohort

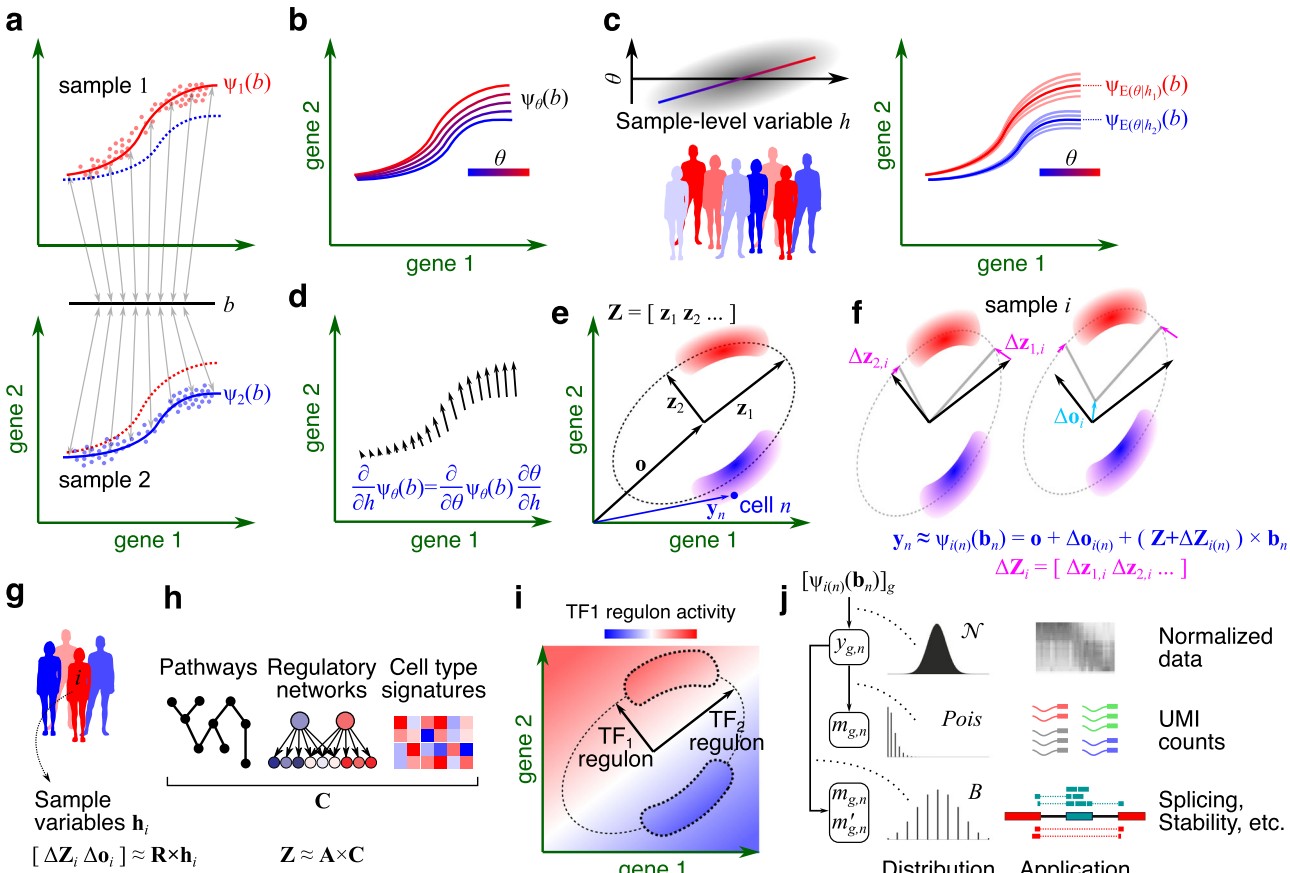

**Fig. 1 | Overview of GEDI. a** Schematic representation of a two-sample single-cell analysis: cells from each sample are distributed near a unique manifold determined by the sample-specific decoder functions $\psi_1$ and $\psi_2$ (each dot represents one cell, with coordinates representing gene expression measurements). These invertible functions provide a mapping (represented by grey arrows) from the biological state of each cell ($b$) to the observed gene expression profile of the cell in each sample. **b** Using a parametric family of functions, the decoder functions can be defined with sample-specific parameters ($\theta$). **c** The sample-specific parameter, $\theta$, can be expressed as a probabilistic function of sample-level variable $h$ (left). Thus, for any given sample characteristic, a distribution of decoder functions can be obtained (right). **d** The derivative of $\psi$ with respect to $h$ forms a vector field, representing the change in expression of each cell at the biological state $b$ as $h$ changes (differential expression). **e** GEDI learns a "reference" manifold in the form of a hyperplane or hyperellipsoid. Here, the manifold is represented by an ellipse, defined by its center

$o$ and principal axes (vectors $z_1$ and $z_2$). Red and blue shaded areas represent cell types and vector $y_n$ represents the expression profile of cell $n$. **f** Transformations of the center and principal axes can distort the reference manifold. Different samples can have different distortions, resulting in non-alignment of cells. **g** GEDI can model the distortions as a linear function of sample-level variables (represented by $h_i$ for each sample $i$). **h** The $z$ vectors in the reference manifold can be modeled as probabilistic functions of prior information $C$. **i** Principal axes aligned to transcription factor (TF) regulons. The color gradient indicates the projected regulon activity of $TF_1$. **j** The GEDI model can be fitted to different types of observations: when the expression of each gene $g$ in each cell $n$ is given (represented by $y_{g,n}$; top), or when $y_{g,n}$ is latent and a probabilistic observation from $y_{g,n}$ is obtained (middle), or a pair of observations representing different event types whose relative proportion is of interest (bottom).

2, the model perfectly distinguishes COVID-19 vs. healthy individuals in cohort 1; conversely, when the model is trained on cohort 1, it achieves an area under receiver operating characteristic (AUROC) curve of 0.97 in cohort 2 (Fig. 2e). Interestingly, similar SVM models trained on pseudobulk-based features did not generalize well across cohorts (Supplementary Fig. 2b-c). Together, these results show that GEDI can capture most of the sample-to-sample variability present in multi-sample scRNA-seq datasets; this variability can then be directly traced back to sample characteristics (such as disease severity) owing to the parametric nature of GEDI's modelling framework.

**GEDI enables cluster-free differential expression analysis**
To enable direct analysis of the relationship between sample-level variables and the gene expression space, we explicitly included them in the GEDI model, by expressing the sample-specific manifold transformations as probabilistic functions of sample-level variables. This model enables us to examine how the manifold and, therefore, the expression vector associated with any given cell state, changes with sample characteristics, providing a transcriptomic vector field for each

sample-level variable (Fig. 3a and Supplementary Fig. 4). We applied this approach to the COVID-19 dataset, to obtain the transcriptomic vector field describing the differences between severe COVID-19 and healthy individuals across the PBMCs. Figure 3b, c provides a visual representation of the cell state space and the transcriptomic vector field of severe COVID-19 over that space. The largest vector magnitudes, corresponding to the cell states that show the largest overall gene expression shift, were observed in plasmablasts, HLA-DR^lo S100A^hi monocytes, and neutrophils (Fig. 3d); the transcriptomic vector magnitudes observed in monocytes and neutrophils recapitulate the previously reported large cell state shifts in these cell types[21].

Calculation of the vector field provides a cluster-free approach for DGE analysis across the continuum of cell states. Nonetheless, it is also possible to perform a traditional cluster-based DGE analysis by summarizing the vector field for any given cell cluster using simple modifications. To showcase this, we calculated the mean COVID-19 transcriptomic vector representing the average shift in gene expression between mild COVID-19 and healthy individuals across all cells of each cell type. Comparison of these cell type-specific mean vectors to

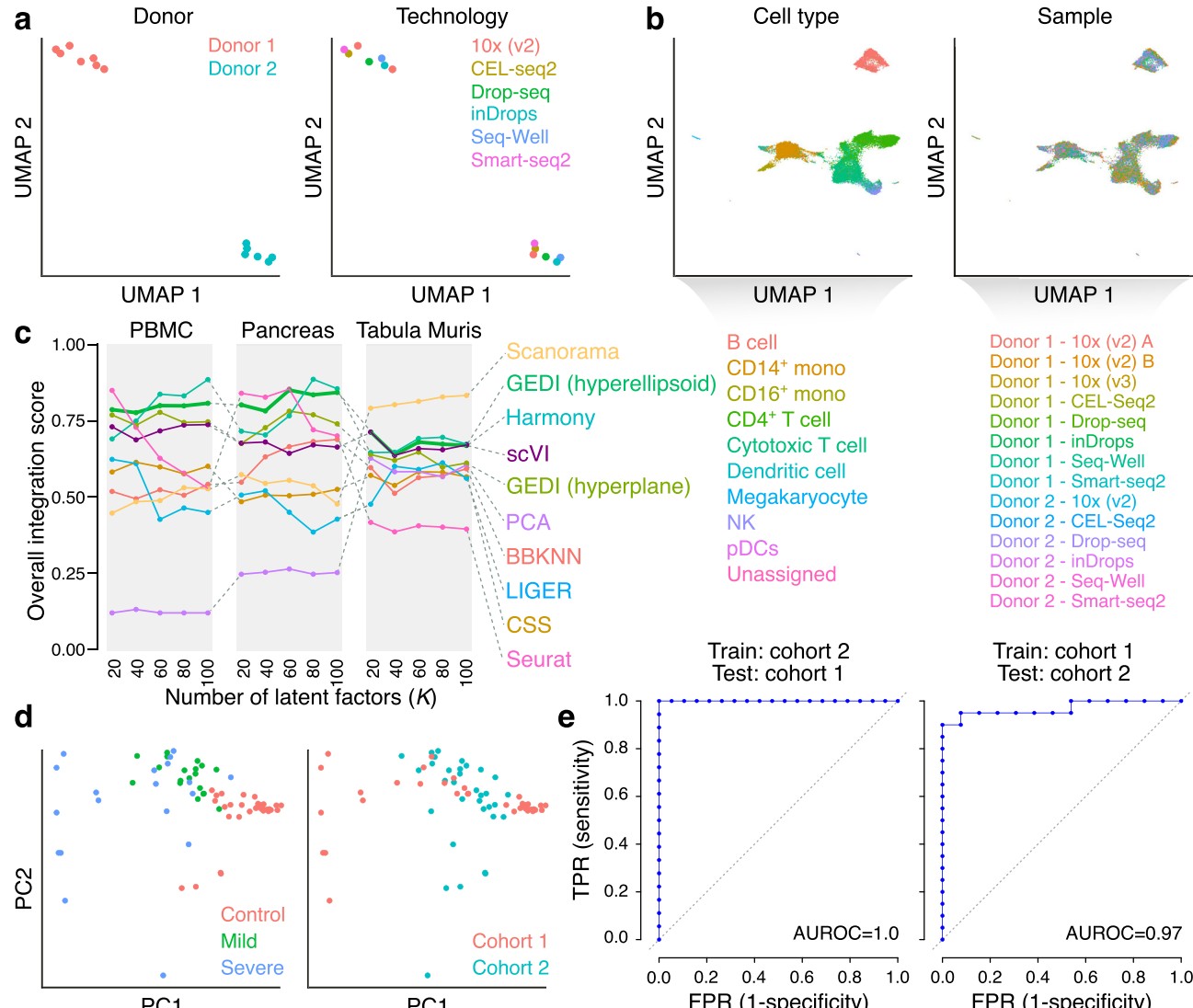

**Fig. 2 | GEDI captures sample-to-sample variability. a** UMAP embedding of the sample-specific manifold distortions learned by GEDI for the PBMC dataset. Each sample was encoded using the set of sample-specific manifold parameters learned by GEDI (excluding sample-specific translation vectors $\Delta o_i$), followed by regressing out the effect of technology from sample-specific parameters *post-hoc*, selection of the top 20 most variable parameters, PCA, and UMAP. Each dot represents one sample, labeled by donor (left) or single-cell technology (right). Only technologies with more than one sample are displayed. See Supplementary Fig. 1a for details and results when the effect of donor is regressed out, and Supplementary Fig. 2a for other choices of top variable features. **b** UMAP embedding of the cells in the PBMC dataset after integration with GEDI ($K = 40$). Each dot represents one cell, colored by the cell type labels from ref. 15 (left) or by sample (right). Also see Supplementary Fig. 1b-f. **c** Overall ranking score comparing the performance of various integration methods over a range of latent factors ($K$), applied to the PBMC,

Pancreas and Tabula Muris datasets. The score reflects the ability to remove technical effects while preserving biological variability, similar to ref. 16 (see **Methods** and Supplementary Data 1 for details, and Supplementary Fig. 3 for additional comparisons). **d** PCA embedding of the sample-specific manifold distortions learned by GEDI for the COVID-19 dataset. Samples were first encoded using the sample-specific parameters, similar to (a), followed by regressing out the effect of cohort and selection of the top 20 most variable parameters for PCA. Each dot represents a sample, labeled by the disease group (left) or the cohort of origin (right). **e** Receiver operating characteristic (ROC) curves assessing the classification between COVID and control cases in the COVID-19 dataset. For the classification task, a Support Vector Machine (SVM) was trained using the top 20 most variable parameters learned by GEDI. Left: SVM was trained with data from cohort 2 and tested on cohort 1. Right: SVM was trained with cohort 1 data and tested on cohort 2. See also Supplementary Fig. 2b. Source data are provided as a Source Data file.

cell type-specific estimates from a pseudo-bulk DGE analysis revealed a high degree of agreement between the two approaches (Fig. 3e and Supplementary Fig. 5a-b). Interestingly, GEDI estimates showed improved reproducibility across cohorts compared to the pseudo-bulk approach (Fig. 3f, g and Supplementary Fig. 5c-d). Furthermore, we found that pseudo-bulk DGE estimates had a substantial correlation between cell types (Supplementary Fig. 6a), whereas GEDI estimates were highly cell type-specific (Supplementary Fig. 6b).

To systematically establish the performance of GEDI in clustering-free DGE analysis, we used GEDI to analyze a simulated cohort-level single-cell dataset, allowing us to compare GEDI's inferences to a

known ground truth for each individual cell. Our simulation framework is schematically shown in Fig. 4a, which is based on a set of synthetic cellular archetypes[22,23] whose expression vectors are determined by sample-level variables, along with additional sources of variation at the sample- and cell-level[5,10]. The parameters needed to simulate cells using this framework can be derived from a variety of sources; we decided to use a real scRNA-seq dataset (specifically, the COVID-19 dataset above) as the template to derive these parameters, in order to preserve characteristics such as gene-gene covariances (see **Methods** for details). We observed that single-cell DE vectors provided by GEDI correlate strongly with the ground-truth DE vector of each cell, with a

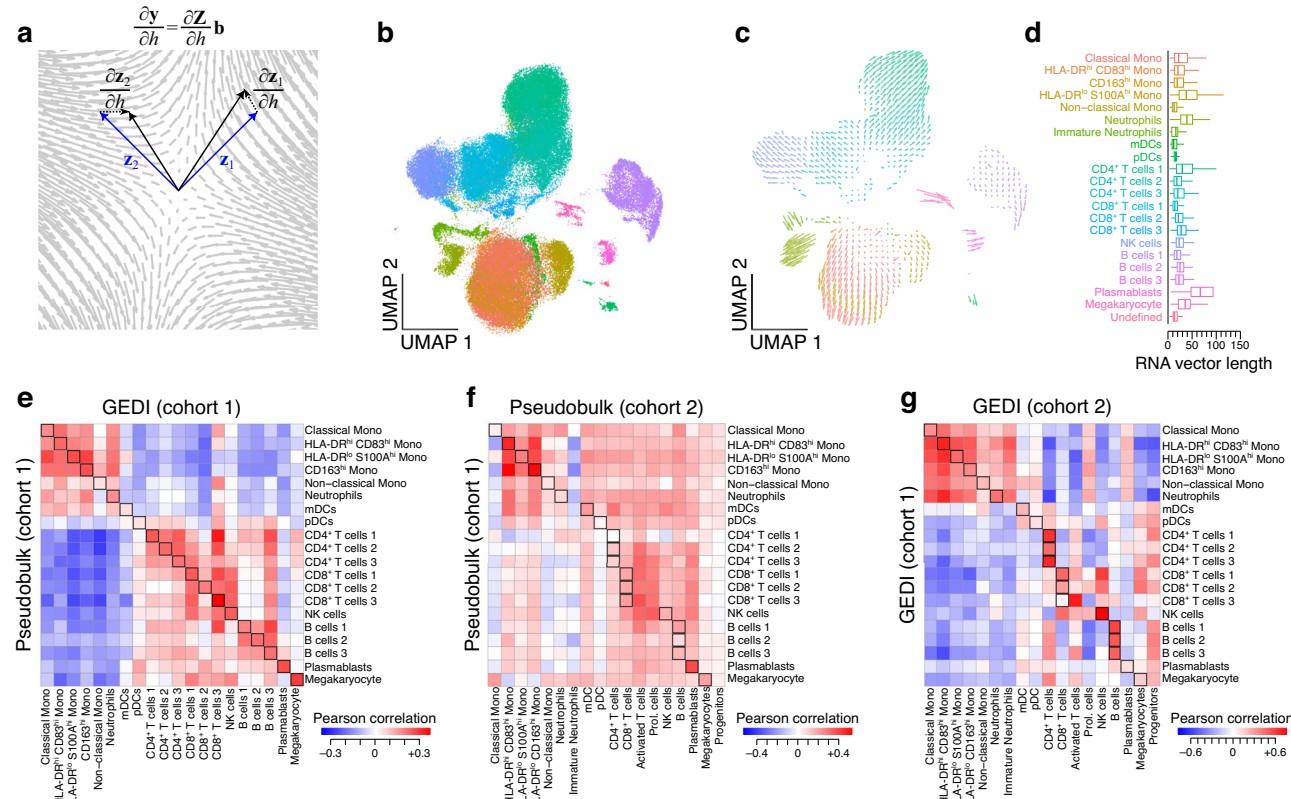

**Fig. 3 | Incorporating sample-level variables in the GEDI model. a** Schematic of a transcriptomic vector field associated with a sample variable *h*. (**b-c**) UMAP embedding of the projection of the transcriptomic vector field of severe COVID-19 for cohort 1. **b** The position of each cell represents the manifold embedding for the control condition—the input for the UMAP was the low-dimensional projection of each cell on the manifold of the healthy group, as learned by GEDI (i.e., the reference manifold plus distortions associated with the control condition). The color indicates the cell type labels from the original study. See Supplementary Fig. 5a for cells colored by donor. **c** Each arrow represents a transcriptomic vector, showing the gene expression change that occurs from the control condition to the severe COVID-19 condition. Arrows are obtained by jointly embedding, in the UMAP space, the (extrapolated) gene expression profiles of each cell in the control and severe COVID-19 condition (see Supplementary Methods for details). **d** Boxplots showing

the magnitude of the gene expression changes (transcriptomic vector magnitude), per cell type, between the control and severe COVID-19 conditions. **e** Comparison between the mean transcriptomic vector field per cell type, obtained from GEDI, and differential gene expression values (log fold-change) obtained from pseudo-bulk analysis. The heatmap shows the Pearson correlation values between the GEDI-based mean vectors (columns) and the pseudobulk-based DE vectors (rows) between cell type pairs, for the comparison of mild COVID-19 vs. control cases in cohort 1 (each element of the heatmap represents Pearson correlation across all genes). **f** Same as in (**e**) but showing reproducibility between cohort 1 (rows) and cohort 2 (columns) for the pseudo-bulk analysis. **g** Same as in **e**−**f** but showing reproducibility between cohort 1 (rows) and cohort 2 (columns) for GEDI. See also Supplementary Fig. 5b-d and Supplementary Fig. 6 for additional comparisons.

slightly better performance when modeling the manifold as a hyper-plane (median Pearson *r* = 0.4, Fig. 4b). In comparison, inferences made by LEMUR[24], another recent method for clustering-free single-cell DE analysis, had significantly lower correlation with ground truth (Mann-Whitney U P < 10[−15]; median *r* = 0.14). At the level of each individual cell, we also defined a set of ground-truth "up-regulated" and "down-regulated" genes by thresholding the ground-truth DE values, and found that GEDI significantly outperforms LEMUR in the identification of up-regulated and down-regulated genes (median AUROC of 0.79 and 0.72 for identification of up-regulated genes by GEDI and LEMUR, and median AUROC of 0.78 and 0.55 for identification of down-regulated genes by GEDI and LEMUR, respectively; all comparisons are significant at P < 10[−15] Fig. 4b). Another recent method, miloDE[25], can also perform clustering-free differential analysis, albeit at the "neighborhood" level as opposed to single-cell level. To compare with miloDE, we collapsed the ground-truth DE vectors as well as GEDI's and LEMUR's inferences to the neighborhood level, by averaging across the cells of each neighborhood (with neighborhoods defined by miloDE). Figure 4c shows that GEDI outperforms both LEMUR and miloDE at the neighborhood level based on different metrics. Finally, we collapsed the ground truth DE vectors as well as GEDI's and LEMUR's inferences to the "cell type" level (see **Methods**),

in order to compare to pseudobulk analysis results obtained by DESeq2. Again, we observed a better agreement between GEDI inferences and the ground truth compared to both LEMUR and DESeq2 (Fig. 4d). These results suggest that GEDI can effectively capture the differential expression of genes at the level of single cells, neighborhoods, and cell types.

## Pathway and network activity projection with GEDI

GEDI can also incorporate prior biological knowledge, such as gene signatures, biological pathways, or GRNs into its model, by expressing the manifold principal axes as probabilistic functions of prior gene-set associations (gene signatures, pathways, and GRNs can be represented by a weighted gene-set association matrix, similar to previous work[8]). As a result, principal axes that can be expressed as a linear combination of one or several gene sets/signatures are deemed more likely by the model, encouraging their alignment with prior knowledge, and allowing the projection of the "activity" of known biological axes onto the cellular states (Fig. 1i). To assess the reliability of these projected activities, we examined GEDI's ability to project cell type signatures across PBMCs. First, using DGE analysis of the PBMC benchmarking dataset, we generated cell-type signatures for each of the two donors (see **Methods** for details). Then, for each donor, we applied GEDI using

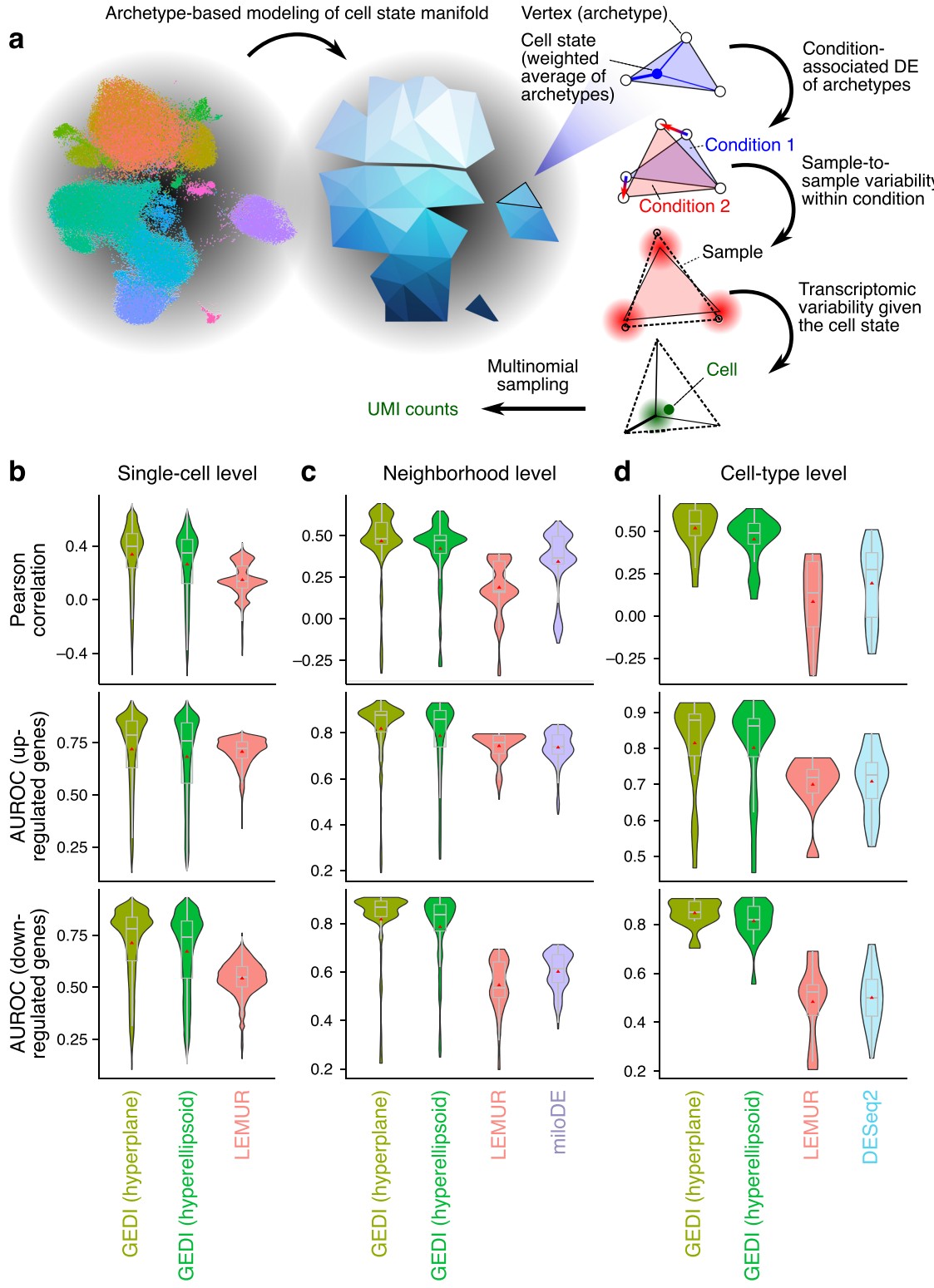

the signatures from the other donor as prior biological knowledge. In both cases, the inferred activity of cell type signatures showed strong enrichment for the true cell type labels (Fig. 5a and Supplementary Fig. 8).

Similarly, when a transcription factor (TF) regulatory network was used as prior biological knowledge (see **Methods**), we observed cell type-specific activity patterns for many TFs (Supplementary Fig. 9), including known lineage regulators such as PAX5 in B cells[26] and TCF7 in CD4+ T cells[27] (Fig. 5b). For 74 out of 89 TFs included in our regulatory network, the projected activity across cells correlated

significantly with the decoded abundance of the mRNA encoding the TF (t-test for Pearson correlation, FDR < 0.001), further supporting GEDI's inferences. The high correlation between the inferred activity of most TFs and their mRNA abundance can also be seen in the COVID-19 data, even when we stratify the cells by their cell type and by the disease condition of the donors (Supplementary Fig. 10). The ability of GEDI to infer TF activities is also supported by its performance on a dataset of single-cell TF perturbations[28]. As shown in Supplementary Fig. 11, for the TFs whose perturbation is associated with cell state shifts along the main axes of variation, GEDI-based activities are highly

**Fig. 4 | Systematic comparison of cluster-free DGE methods using a simulated cohort-level single-cell dataset. a** Schematic representation of our framework to simulate cohort-level scRNA-seq data. We start by simulating the cell state manifold of each sample as a set of "archetypes" while constraining any cell state to some weighted average of those archetypes; thus, the cell states are confined within the polytope defined by the archetypes. In each sample, the gene expression vector of each archetype is determined by the sample-level variables (plus an additional variance that is not explained by sample characteristics), resulting in a ground-truth DE vector for each archetype and, by extension, any given cell state. Finally, each cell is drawn from a distribution centered on a given cell state in a given sample, followed by simulating UMI counts (see **Methods** for more details). **b** Single-cell-level performance on a simulated dataset. The simulation parameters were

extracted from comparison of mild COVID-19 vs. control individuals (see Supplementary Methods). For each cell (n of cells = 86,549), the differential expression estimates were compared against the ground truth via Pearson correlation (top), or by assessing the classification of up-regulated (middle) or down-regulated (bottom) genes using AUROC. Sets of up-regulated and down-regulated genes were defined using a threshold of 0.3 on the $\log_2$ ground-truth DE values. For each metric, the violin plots and boxplots show the distribution across all cells; red triangle: mean, center line: median; box limits: upper and lower quartiles; whiskers: 1.5x the interquartile range. **c** Same as in **b** but at the neighborhood level (n of neighborhoods = 215). **d** Same as in **b** but at the cell type level (n of cell types = 20). See also Supplementary Fig. 7 for simulations based on comparison of severe COVID-19 vs. control.

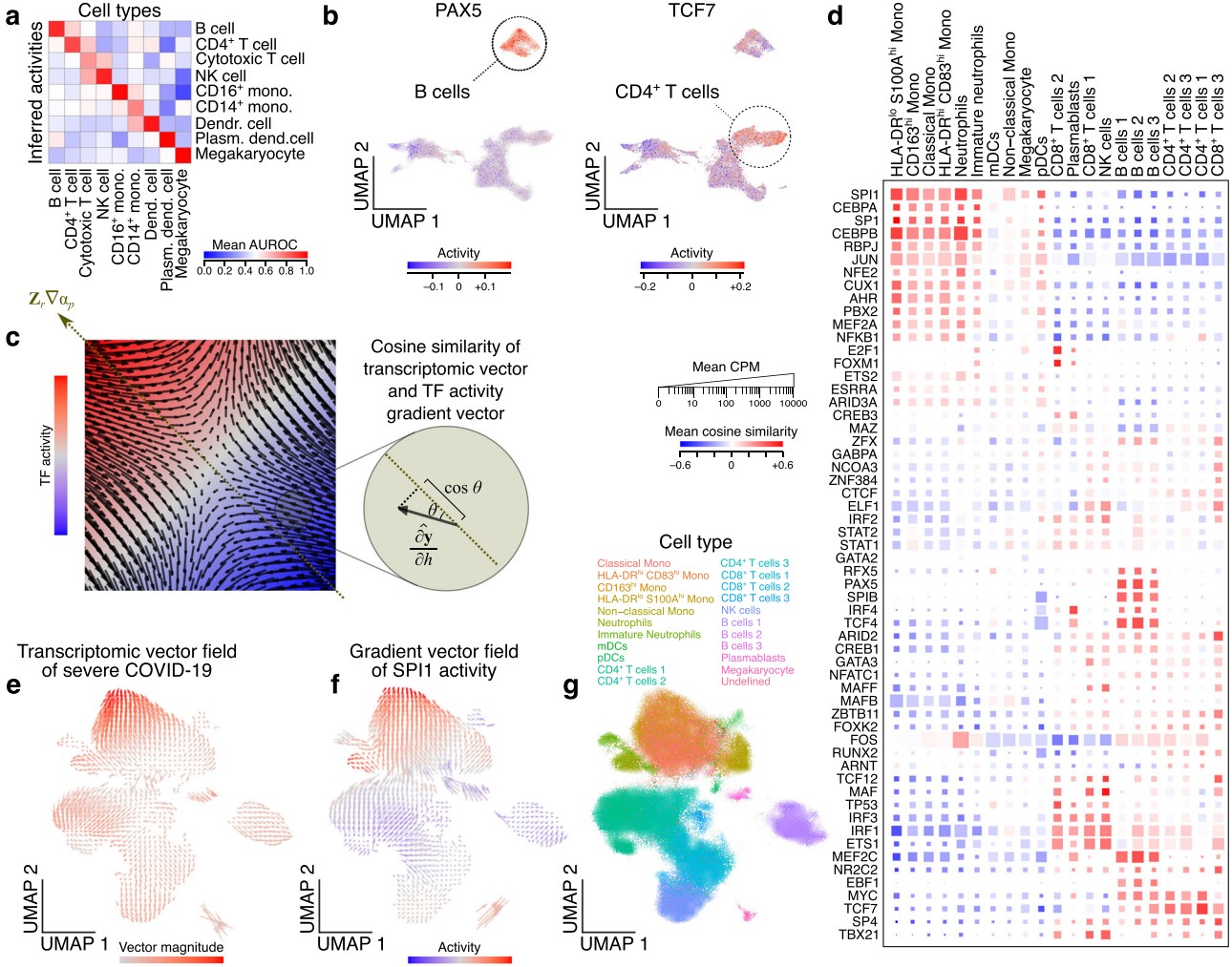

**Fig. 5 | Modeling the manifold as a function of gene-level variables. a** Cell type signature projections obtained by GEDI are compared to the true labels in the PBMC dataset (donor 1). Heatmap shows AUROC values for differential enrichment of inferred cell type signatures from GEDI (rows) for each cell type (columns). See **Methods** for details. Also see Supplementary Fig. 8. **b** Examples showing single-cell projection of transcription factor (TF) regulon activities. Each UMAP shows the cells from the PBMC dataset, with the regulon activity of PAX5 (left) and TCF7 (right) shown using the color gradient. The UMAP embedding is identical to that shown in Fig. 2b. The cell type with the highest activity for each TF is highlighted. See Supplementary Fig. 9 and Supplementary Data 2 for cell type-specific activities of other TFs. **c** Schematic illustration of TF regulon activity gradient and its relationship to transcriptomic vector field associated with a sample variable. The color shows the regulon activity, with its gradient represented by the diagonal vector. The arrows within the square represent the vector field of sample-level variable *h*. The inset

shows how the cosine similarity of a transcriptomic vector and the TF regulon activity vector can be obtained. **d** Comparison between the TF gradient vector field (rows) and the transcriptomic vector field of severe COVID-19 per cell type (columns). In each square, the color represents the mean cosine similarity of the COVID-19 vector and the TF gradient vector. The square size represents the mean CPM, per cell type, of the mRNA encoding each TF. Only TFs with Pearson correlation ≥0.25 between their inferred activity and their mRNA abundance are shown. **e-g** An example TF whose activity gradient correlates with the transcriptomic vector field of severe COVID-19 in monocytes. **e** UMAP representation of the transcriptomic vector field of severe COVID-19. The color shows the vector magnitude. (**f**) Gradient vector field of SPI1 activity. The color represents SPI1 activity. **g** The same UMAP as in (e-f), but the color represents the cell type labels. See Supplementary Fig. 12 for other examples. Source data are provided as a Source Data file.

predictive of the TF perturbation status. For this subset of TFs, GEDI (hyperellipsoid) is in fact among the top performers compared to eight other methods we tested, with mean AUROC of 0.86 for distinguishing the cells in which a specific TF is perturbed from other cells. Given that GEDI only models the principal axes of variation as functions of the gene regulatory network, its behaviour in correctly modeling the TFs that cause cell state shifts along these axes is expected.

GEDI network activity projection also enables the calculation of a gradient vector for the activity of each TF, representing the direction of greatest increase in TF activity in the cell space (Fig. 5c). One can then compare the gradient vector of each TF to a given transcriptomic vector field, to examine whether in certain cellular states the transcriptomic vector field is aligned with the gradient vector of that TF (Fig. 5c). We used GEDI to infer the regulon activity of TFs in the COVID-19 dataset and, for each single cell, compared the activity gradient vector of each TF to the transcriptomic vector field of severe COVID-19. Interestingly, the transcriptomic vector field of severe COVID-19 correlated (or anti-correlated) strongly with the TF activity gradients in a cell type-specific pattern (Fig. 5d). In other words, in certain cell states, when we move from healthy to severe COVID-19, the direction of change in the gene expression coincides with the direction of greatest increase (or decrease) in the activity of specific TFs. For example, the activity gradient of a group of TFs showed high concordance with the transcriptomic vector field of severe COVID-19 in HLA-DR$^{lo}$ S100A$^{hi}$ monocytes, including SPI1, CEBPA, and SP1 (Fig. 5e–g).

and Supplementary Fig. 12), suggesting that severe COVID-19 is accompanied by increased activity of these TFs in HLA-DR$^{lo}$ S100A$^{hi}$ monocytes. Among these TFs, we observed strong up-regulation for *SPI1* mRNA in monocytes in severe COVID-19 compared to healthy controls (pseudobulk DE and GEDI cluster-free DE analyses; Supplementary Fig. 13a), but not for the other two TFs. Nonetheless, for all three TFs, the direct targets whose expression was highly correlated with the expression of the TF were enriched in various immune-related pathways (Supplementary Fig. 13b).

## Modeling the latent space of RNA splicing and stability with GEDI

In contrast to the analysis of mRNA abundance, where for each cell and each feature a single quantity is recorded (e.g., the UMI count), analysis of many other biological processes requires working with the ratio of two quantities. For example, analysis of alternative splicing of cassette exons involves modeling the percent-spliced-in (PSI), representing the ratio between the abundances of isoforms in which the cassette exon is included vs. excluded[11] (see Supplementary Fig. 14 for other examples). Such analysis is further complicated, especially in single-cell data, by the fact that the quantities whose ratio is of interest are latent, and instead some probabilistic observation is obtained (e.g., UMI counts of inclusion or exclusion isoforms). By using a hierarchical model in which the latent profile of each cell is connected to the observed data through a binomial data-generating distribution (Fig. 6a), GEDI

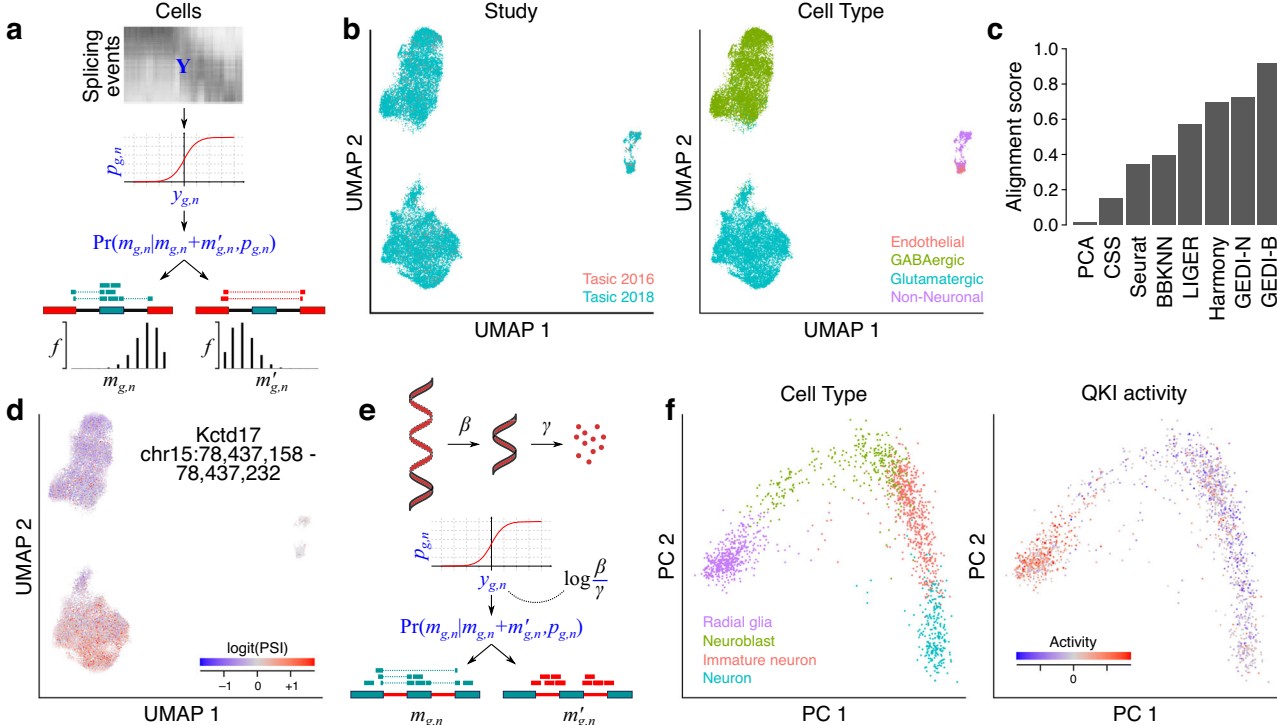

**Fig. 6 | Modeling the latent space of RNA splicing and stability with GEDI. a** Schematic representation of modeling exon inclusion levels. The log-odds of inclusion/exclusion of exon *g* in cell *n* ($y_{g,n}$), which is unobserved (latent), is connected to the observed counts of reads that support the inclusion of that exon ($m_{g,n}$) or its exclusion ($m'_{g,n}$) through a binomial distribution; the distribution parameter $p_{g,n}$ represents the percent-spliced-in (PSI). GEDI models the latent space of exon inclusion/exclusion across cells and samples similar to Fig. 1. **b** UMAP embedding of the latent splicing space of mouse cortical cells after integration of data from two studies[29,30]. Labels represent the study of origin (left) or the cell type labels from the original study (right). **c** Alignment score for removal of the technical variation between the two datasets using different methods. GEDI was applied either assuming a binomial distribution (B) or normal distribution (N) for the input data. For all methods excluding GEDI-B, a naïve estimate of logit (PSI) was used as

input (See **Methods**), whereas GEDI-B was directly fitted to inclusion/exclusion counts following the model shown in **a**. **d** An example cassette exon that is differentially spliced between neuronal subtypes. The UMAP is the same as in (b), but the color represents the log-odds of inclusion/exclusion for a cassette exon in *Kctd17*. See also Supplementary Fig. 15. **e** A similar model can be fitted to the counts of exonic and intronic reads, with the resulting latent space representing the log-ratio of pre-mRNA processing rate (*β*) to mRNA degradation rate (*γ*). Assuming invariability of processing rate, this ratio can be interpreted as mRNA stability at steady state[12]. **f** Modeling the mRNA stability manifold as a function of RNA-binding protein (RBP) regulons. The color shows cell type labels (left) or the projected regulon activity of QKI (right). See Supplementary Fig. 17 for other examples. Source data are provided as a Source Data file.

extends the analyses described in the previous sections to paired quantities whose ratio is of interest.

We applied GEDI to the analysis of exon inclusion levels in the mouse cortex using data from two previously published studies[29,30]. We observed that the latent splicing space learned by GEDI, which represents the lower-dimensional projection of the cells based on their (unobserved) cassette exon PSI values, preserved the cell type structure, while removing the study-specific effects (Fig. 6b). We then compared the ability of GEDI to integrate the latent splicing space of multiple samples against that of other integration methods—note that, to apply other methods, a naïve estimate of PSI needed to be calculated first, while GEDI could be directly fitted to inclusion/exclusion counts. We found that GEDI offered the best performance at removing technical variability (Fig. 6c). Furthermore, as part of its expectation-maximization algorithm for model fitting, GEDI calculates the expected value of the (latent) ratio of inclusion/exclusion events given the observed counts and the model parameters, effectively providing a denoised estimate of PSI. We found that GEDI-inferred PSI values recapitulated previously observed cell type-specific splicing events, e.g., the inclusion of exon 20 of *Nrxn1* in GABAergic neurons and exon 2 of *Ntpn* in glutamatergic neurons[31] (Supplementary Fig. 15a), as well as other differentially spliced exons that are enriched for neuron-related functions (Supplementary Fig. 15b-c). It also identified novel associations between cassette exons and neuronal subtypes, such as the glutamatergic neuron-specific inclusion of a cassette exon in *Kctd17* (Fig. 6d and Supplementary Fig. 15d). This observation is consistent with simulations showing that GEDI can impute ground truth ratios from paired, sparse counts, while a naïve estimator provides ratios that are almost completely uncorrelated with the ground truth (Supplementary Fig. 16a-e). Using the naïve estimator as input for two existing single-cell imputation methods[2,32] slightly improved its correlation with the ground truth, but GEDI substantially outperformed them in recovering the ground truth (Supplementary Fig. 16f-g).

Finally, we evaluated the ability of GEDI to model RNA stability based on the ratio of spliced and unspliced RNA, assuming that RNA stability is proportional to the spliced/unspliced transcript ratio at steady-state in the absence of changes in RNA processing rate[12] (Fig. 6e). While these conditions are not fully met in every single cell, we reasoned that spliced/unspliced transcripts ratios are still informative of RNA stability in cells that are not in the middle of a differentiation trajectory (and, therefore, are closer to steady state). To test this hypothesis, we applied GEDI to analyze the spliced/unspliced ratio of RNAs at the single-cell level in a model of sensory neurogenesis[33]. We compared the log-ratio of spliced vs. unspliced transcripts, as imputed by GEDI, to RNA half-life measurements obtained from mouse embryonic stem cells (ESCs) and in vitro-differentiated terminal neurons (TNs)[12]. Despite the differences between the biological systems represented by these two datasets, we observed a Pearson correlation of 0.16 between GEDI inferences and differential mRNA half-life measurements (Supplementary Fig. 17a), compared to a Pearson correlation of 0.22 when bulk RNA-seq data from the same in vitro differentiation system is used[34], or to a mean Pearson correlation of 0.002 for shuffled single-cell data (Supplementary Fig. 17b). We then used GEDI to analyze spliced/unspliced transcript ratios in human neurons, using a previously published dataset of human embryonic glutamatergic neurogenesis[13]. In this analysis, we modeled the spliced/unspliced manifold as a function of the regulatory networks of RNA binding proteins and miRNAs (see **Methods** for details). GEDI was able to recover cell type-specific activities of several known post-transcriptional regulators (Supplementary Fig. 17c), including a higher projected activity of well-characterized factors such as QKI in radial glia[35] (Fig. 6f) and miR-124 in differentiated neurons[36] (Supplementary Fig. 17d). Collectively, these results show that GEDI can successfully model the latent space of RNA splicing and stability at the single-cell level.

## Discussion

GEDI is a specific formulation of a general framework for multi-sample single-cell analysis (Fig. 1a–d) in which a family of functions, parametrized by sample-specific factors, connect each biological cell state to its expected expression profile in each sample. The framework proposed here includes three key components. First, it requires the identification of invertible decoder functions that provide a map from gene expression space to cell state manifold and vice versa, allowing the development of generative models of single-cell data. This requirement distinguishes this framework from unsupervised manifold alignment problem[37] and the majority of existing single-cell data integration approaches, such as those based on correlation analysis[3,38] or graph-based methods[4] in which a given gene expression profile can be mapped to its cell state but not vice versa, or methods based on autoencoders[39] in which the encoder and decoder functions are not necessarily the inverse of each other.

Secondly, in the hierarchical model proposed here, sample-specific manifolds are drawn from a distribution around a mean manifold, with the mean manifold optionally expressed as a function of sample covariates. This probabilistic modeling of the manifold, which was inspired by the methodology used by LIGER[40] for inter-sample harmonization, separates our framework from LEMUR, another recent method for latent embedding regression[24], in which the latent space is a deterministic function of sample covariates without accounting for variation among biological replicates. Previous work has shown that properly modeling sample-to-sample variability is key for unbiased differential expression analysis in single-cell data[5,10], which may partially explain the superior performance of GEDI in our simulation-based benchmarking tests (Fig. 4). We note, however, that more extensive analyses are needed to better understand the effects of factors that may influence the performance of GEDI and other DE analysis methods, including the number of differentially expressed genes per cell, the magnitude of their differential expression, cell-cell and gene-gene correlation structures, inter- and intra-sample variances, the number of samples, the number of cells per sample, and the sequencing depth.

Third, the hierarchical framework proposed here models each cell as a sample drawn from a distribution around the manifold of each sample, followed by probabilistic sampling of the observed data from the latent gene expression profile of the cell. This hierarchical structure allows the same model to be generalized to different observation types by employing different data-generating distributions, and underlies the distinguishing feature of GEDI to not only model the latent space of mRNA expression, but also the stability and splicing latent spaces of single cells. We expect this functionality to be useful for analysis of other single-cell modalities that represent the ratio of two biological measurements, as summarized in Supplementary Fig. 14, enabling a range of analyses based on those modalities, including batch correction and cluster/cell type analysis (similar to Fig. 6a–c). This hierarchical model also provides a natural method for denoising and/or imputation of single-cell data: we treat the true profile of each cell as a latent variable, the expected value of which can be calculated conditional on the observed data and the maximum *a posteriori* estimate of the model parameters (examples can be found in Fig. 6d and Supplementary Fig. 15). As shown in Supplementary Fig. 16, our simulation results underline the unique ability of GEDI to impute the ratios of paired observations (such as spliced vs. unspliced mRNA abundances), but it remains to be tested whether GEDI's imputations for gene-level expression values are also competitive with existing single-cell imputation methods.

In addition, our formulation enables direct modeling of the parameters of the decoder (and, therefore, the manifold defined by that decoder) as functions of prior knowledge in the form of gene sets. This direct incorporation of prior biological knowledge into the GEDI model, which was inspired by previous work on latent variable modeling in bulk expression data[8], provides several unique advantages.

First, GEDI penalizes principal axes that cannot be expressed as a linear combination of gene-level prior knowledge, facilitating the downstream interpretability of the resulting manifold axes. Secondly, gene set activities can be projected onto individual cells, thus providing a natural approach to perform single-cell gene set enrichment analysis. This flexible framework enables the study of different types of regulatory mechanisms depending on the observations modelled. For example, it can be employed for the study of transcriptional regulators if gene-level counts are given as input (e.g., Supplementary Fig. 9), or for the analysis of regulatory networks that modulate mRNA stability if paired spliced and unspliced transcript counts are provided (e.g., Supplementary Fig. 17c). Thirdly, the activity gradient of each gene set can be directly compared to any transcriptomic vector field, such as the vector field associated with a specific sample-level feature, ultimately associating the activity of known gene sets with gene expression shifts observed across conditions (Fig. 5d–g).

We envision several potential extensions of this work. First, as noted earlier, our framework can be extended to different parametric manifold approximation approaches (GEDI currently supports linear manifold learning, with the option to further restrict the manifold to a hyperellipsoid). Secondly, it provides a natural path for extension of its concepts to multi-modal mosaic integration[40]: by using a different family of decoder functions for each modality, we can obtain simultaneous mapping from any given biological state to the spaces of different modalities. The proposed generative model can also naturally handle missing data, simply by excluding missing observations from the calculation of the *a posteriori* probability. Third, this framework can be used not only for de novo modeling of sample-specific manifolds, but also for post-hoc analysis of the harmonized space obtained by other existing methods. To showcase this potential, we used the integrated space identified by Harmony[41] for the samples in cohort 1 of the COVID-19 dataset, and asked GEDI to identify sample-specific transformations that can approximate Harmony's integrated space, model those sample-specific transformations as a function of disease status, and calculate the transcriptomic vector field of mild COVID-19 relative to healthy controls, as shown in Supplementary Fig. 18. Fourth, the transcriptomic vector fields obtained by GEDI may have applications beyond clustering-free DE analysis. For example, earlier studies[42] have shown the utility of transcriptomic vector fields in prediction of cell fate transitions if the vector field represents "velocity" (gene expression change as a function of time). Furthermore, as GEDI's vector field extends beyond the regions of the manifold that is occupied by observed cells, it may provide an opportunity for counterfactual prediction in previously unobserved cell types, such as prediction of response to specific perturbations[43,44]. These potential abilities, however, remain currently untested.

Overall, the framework presented here unifies a range of concepts that are central to single-cell data analysis, including multi-sample data integration, cluster-free DGE analysis, imputation and denoising, pathway and GRN activity analysis using prior information, downstream model interpretation, and analysis of different modalities with distinct data-generating processes.

## Methods

### The GEDI framework

Consider a dataset with measurements for $G$ genes/events in $N$ cells. Let the column vector $\mathbf{y}_n \in \mathbb{R}^G$ denote the expression values for $G$ genes in cell $n \in \{1,...,N\}$. The vectors $\mathbf{y}_n$ ($n \in \{1,...,N\}$) together form the matrix $\mathbf{Y} \in \mathbb{R}^{G \times N}$, in which each column $n$ can be considered an observation in a $G$-dimensional space. We further assume that these observations lie near a lower-dimensional manifold, so that some function $\psi$ can reconstruct each $\mathbf{y}_n$ from a lower-dimensional column vector $\mathbf{b}_n \in \mathbb{R}^K$ ($K < G$):

$$\psi_\theta : \mathbb{R}^K \to \mathbb{R}^G \tag{1}$$

$$\mathbf{y}_n \cong \psi_\theta(\mathbf{b}_n) \tag{2}$$

Here, $\theta$ is the set of parameters that define the manifold, and $\mathbf{b}_n$ represents the embedding of the $n$'th observation on this manifold.

Furthermore, the $N$ cells in the dataset may belong to different samples. Each sample $i \in \{1,...,Q\}$ may have a different (distorted) manifold, defined by the parameter set $\theta_i = \theta_r + \Delta\theta_i$. Therefore, when the cells are derived from multiple samples, each observation $\mathbf{y}_n$ can be modeled as:

$$\theta = \theta_r + \triangle\theta_{i(n)} \tag{3}$$

$$\mathbf{y}_n \cong \psi_\theta(\mathbf{b}_n) \tag{4}$$

Here, $\Delta\theta_{i(n)}$ represents the difference between $\theta_r$, the parameter set that defines a "reference" manifold, and $\theta_{i(n)}$, the parameter set that defines the manifold of the sample to which cell $n$ belongs (we denote the sample from which the cell $n$ is derived as $i(n)$). This formulation allows direct mapping between the cells (each defined by a constant embedding $\mathbf{b}$) across multiple samples (through sample-specific manifold parameterization).

The general concept above can potentially be adapted to various parametric manifold learning methods; GEDI represents a specific case, in which the gene expression manifold is modeled as a $K$-dimensional hyperplane (with the option to further restrict the manifold shape, as described later). This is achieved by defining the function $\psi$ as:

$$\theta = \{\mathbf{o}, \mathbf{Z}\} \tag{5}$$

$$\psi_\theta(\mathbf{b}_n) = \mathbf{o} + \mathbf{Z}\mathbf{b}_n \tag{6}$$

In a multi-sample analysis, sample-specific parameter sets are defined as:

$$\theta_i = \{\mathbf{o}_r + \triangle\mathbf{o}_i, \mathbf{Z}_r + \triangle\mathbf{Z}_i\} \tag{7}$$

$$\Theta = \{\theta_1, \ldots, \theta_Q\} \tag{8}$$

Here, the column vector $\mathbf{o}_r \in \mathbb{R}^G$ represents the origin point (center) on a reference hyperplane, and $\Delta\mathbf{o}_i$ represents the sample-specific translation of the origin point. Each of the columns of the matrix $\mathbf{Z}_r \in \mathbb{R}^{G \times K}$ represents a vector that originates from point $\mathbf{o}_r$ and lies on the reference hyperplane. By default, GEDI restricts these $K$ vectors to be orthogonal to each other, effectively forming the orthogonal axes of a coordinate frame in which the position of each point on the reference manifold can be specified as $\mathbf{b}_n$. $\Delta\mathbf{Z}_i$ represents the sample-specific transformations of this coordinate frame (excluding translation, which is specified by $\Delta\mathbf{o}_i$).

Thus, GEDI approximates each observation $\mathbf{y}_n$ as:

$$\mathbf{y}_n \cong \mathbf{o}_r + \triangle\mathbf{o}_{i(n)} + \left(\mathbf{Z}_r + \triangle\mathbf{Z}_{i(n)}\right)\mathbf{b}_n \tag{9}$$

More precisely, $\mathbf{y}_n$ is modeled as an observation drawn from a spherical multivariate normal distribution whose mean is located on the manifold of sample $i(n)$:

$$\mathbf{y}_n | \boldsymbol{\mu}_n(\Theta) \sim \mathcal{N}\left(\boldsymbol{\mu}_n(\Theta), \sigma^2 \mathbf{I}\right) \tag{10}$$

$$\boldsymbol{\mu}_n(\Theta) = \mathbf{o}_r + \triangle\mathbf{o}_{i(n)} + \left(\mathbf{Z}_r + \triangle\mathbf{Z}_{i(n)}\right)\mathbf{b}_n + s_n\mathbf{1}_G \tag{11}$$

Note the addition of the term $s_n\mathbf{1}_G$ here, which serves as a cell-specific intercept; $s_n$ is a scalar (representing library size), and $\mathbf{1}_G$ is a column vector of 1's; $\mathbf{1}_G = \{1,...,1\} \in \mathbb{R}^G$.

The column vectors $\mathbf{b}_{n \in \{1,\ldots,N\}}$ together form the matrix $\mathbf{B} \in \mathbb{R}^{K \times N}$, in which each column $n$ can be considered the embedding of the cell $n$ in the manifold. Since the scales of $\mathbf{Z}_r + \Delta\mathbf{Z}_i$ and $\mathbf{B}$ are redundant (each column of $\mathbf{Z}_r + \Delta\mathbf{Z}_i$ can be scaled by some constant $c$ and the corresponding row of $\mathbf{B}$ can be scaled by $c^{-1}$ without changing the model likelihood), GEDI restricts $\mathbf{B}$ such that each row forms a unit vector:

$$\mathbf{B} \in \left\{ \mathbb{R}^{K \times N}, |, \forall k \in \{1,\ldots,K\} \sum_{n=1}^{N} (b_{k,n})^2 = 1 \right\} \quad (12)$$

Other constraints may also be added to further limit the shape of the manifold. For example, *B* may be restricted to the points on a ellipsoid; in other words:

$$\mathbf{B} \in \left\{ \mathbb{R}^{K \times N}, |, \forall k \in \{1,\ldots,K\} \sum_{n=1}^{N} (b_{k,n})^2 = 1 \right\}$$
$$\cap \left\{ \mathbb{R}^{K \times N}, |, \exists \mathbf{d} \mathbb{R}_{>0}^K \; s.t. \; \forall n \in \{1,\ldots,N\} \sum_{k=1}^{K} (b_{k,n}/d_k)^2 = 1 \right\} \quad (13)$$

Here, the vector $\mathbf{d}$ contains the lengths of the semi-axes of the ellipsoid, with $d_k$ representing the $k$th element of $\mathbf{d}$.

**Modeling the reference manifold as a function of gene-level variables.** The reference manifold itself may be approximated as a function of gene-level prior knowledge, such as gene regulatory networks or pathway memberships, by expressing the parameter set $\theta_r$ as a function of gene-level variables. To this end, GEDI expresses $\mathbf{Z}_r$ as a probabilistic function of $\mathbf{C} \in \mathbb{R}^{G \times P}$, where $\mathbf{C}$ is a matrix representing gene-level prior information matrix:

$$\mathbf{z}_{r,k} | \mathbf{a}_k \sim \mathcal{N} \left( \mathbf{C}\mathbf{a}_k, \sigma^2 S_Z \mathbf{I} \right) \quad (14)$$

$$\mathbf{a}_k \sim \mathcal{N} \left( \mathbf{0}, \sigma^2 S_A \mathbf{I} \right) \quad (15)$$

Here, $\mathbf{z}_{r,k} \in \mathbb{R}^G$ is the $k$'th column of $\mathbf{Z}_r$, and $\mathbf{a}_k \in \mathbb{R}^P$ is a vector column whose $P$ elements represent the contribution of each of the $P$ columns of $\mathbf{C}$ toward determining the direction of the axis vector $\mathbf{z}_{r,k}$—this formulation is similar to (and inspired by) that used by PLIER[8] for pathway-level information extraction from gene expression data. $S_Z$ is a hyperparameter that determines the variance of $\mathbf{z}_{r,k}$ relative to the model variance $\sigma^2$. Similarly, $S_A$ is a hyperparameter that determines the variance of the prior distribution of $\mathbf{a}_k$ relative to the model variance $\sigma^2$ (see Supplementary Methods for the choice of $S_Z$, $S_A$, and other hyperparameters).

The column vectors $\mathbf{a}_{k \in \{1,\ldots,K\}}$ together form the matrix $\mathbf{A} \in \mathbb{R}^{P \times K}$. For example, $\mathbf{C}$ may represent a (weighted) regulatory network connecting $P$ transcription factors to $G$ genes, in which case the element $a_{p,k}$ of the matrix $\mathbf{A}$ corresponds to the contribution of the network of the transcription factor $p$ toward the axis vector $\mathbf{z}_{r,k}$. $\mathbf{A}\mathbf{b}_n$ can be considered as the projected "activity" of the $P$ transcription factors in the cell $n$ ($\mathbf{A}\mathbf{b}_n \in \mathbb{R}^P$). In other words, $\alpha_p(\mathbf{b}) = \mathbf{a}_{p,\cdot}\mathbf{b}$ is the function that provides the projected activity of the transcription factor $p$ at coordinate $\mathbf{b}$ in the $\mathbf{Z}_r$ coordinate system, where $\mathbf{a}_{p,\cdot}$ is $p$'th row of $\mathbf{A}$. Consequently, the gradient of the activity of transcript factor $p$ in the $\mathbf{Z}_r$ coordinate system ($\nabla\alpha_p$) is $\mathbf{a}_{p,\cdot}^\top$, which can be transformed to the gene expression coordinate system as $\mathbf{Z}_r \nabla\alpha_p = \mathbf{Z}_r \mathbf{a}_{p,\cdot}^\top$.

In the absence of gene-level prior information, the following prior is used for $\mathbf{z}_{r,k}$:

$$\mathbf{z}_{r,k} \sim \mathcal{N} \left( \mathbf{0}, \sigma^2 S_Z \mathbf{I} \right) \quad (16)$$

**Modeling sample-specific distortions of the manifold as a function of sample-level variables.** The difference between the manifold of each sample $i$ and the reference manifold can be expressed using the difference of the parameter sets that define these manifolds, i.e., $\Delta\theta_i$. In the case of GEDI, we have: $\Delta\theta_i = \{\Delta\mathbf{o}_i, \Delta\mathbf{z}_{i,1}, \ldots, \Delta\mathbf{z}_{i,K}\}$. Each of the components can in turn be expressed as a function of sample-level variables:

$$\mathbf{o}_i | \mathbf{R}_o \sim \mathcal{N} \left( \mathbf{R}_o \mathbf{h}_i, \sigma^2 S_{\Delta o_i} \mathbf{I} \right) \quad (17)$$

$$\mathbf{R}_o \sim \mathcal{N} \left( \mathbf{0}, \sigma^2 S_{R_o} \mathbf{I} \right) \quad (18)$$

$$\mathbf{z}_{i,k} | \mathbf{R}_k \sim \mathcal{N} \left( \mathbf{R}_k \mathbf{h}_i, \sigma^2 S_{\Delta Z_i} \mathbf{I} \right) \quad (19)$$

$$\mathbf{R}_k \sim \mathcal{N} \left( \mathbf{0}, \sigma^2 S_{R_k} \mathbf{I} \right) \quad (20)$$

Here, $\mathbf{h}_i \in \mathbb{R}^L$ is a column vector whose elements represent the values of $L$ variables for sample $i$. $\mathbf{R}_o \in \mathbb{R}^{G \times L}$ and $\mathbf{R}_k \in \mathbb{R}^{G \times L}$ are matrices that represent the effects of the $L$ variables on $\Delta\mathbf{o}_i$ and $\Delta\mathbf{z}_{i,k}$, respectively. $S_{\Delta oi}$ and $S_{\Delta zi}$ are sample-specific hyperparameters that specify the variance of the $\Delta\mathbf{o}_i$ and $\Delta\mathbf{z}_{i,k}$ relative to the model variance $\sigma^2$. Similarly, $S_{Ro}$ and $S_{Rk}$ are hyperparameters that determine the variance of the prior distributions of $\mathbf{R}_o$ and $\mathbf{R}_k$ relative to the model variance $\sigma^2$.

In the absence of sample-level variables, $\Delta\mathbf{o}_i$ and $\Delta\mathbf{z}_{i,k}$ are modeled using the following prior distributions:

$$\mathbf{o}_i \sim \mathcal{N} \left( \mathbf{0}, \sigma^2 S_{\Delta o_i} \mathbf{I} \right) \quad (21)$$

$$\mathbf{z}_{i,k} \sim \mathcal{N} \left( \mathbf{0}, \sigma^2 S_{\Delta Z_i} \mathbf{I} \right) \quad (22)$$

**Direct inference from count data.** Consider the count matrix $\mathbf{M} \in \mathbb{Z}^{G \times N}$, with each element $m_{g,n}$ generated from a Poisson distribution with mean $\lambda_{g,n}$. GEDI models each $\lambda_{g,n}$ as a latent variable drawn from a log-normal distribution, so that $\mathbf{y}_n = (\log\lambda_{1,n}, \ldots, \log\lambda_{G,n})^\top \in \mathbb{R}^G$ follows a spherical multivariate normal distribution as in the previous sections. Thus:

$$m_{g,n} \sim \text{Pois}(e^{y_{g,n}}) \quad (23)$$

$$\mathbf{y}_n | \boldsymbol{\mu}_n(\Theta) \sim \mathcal{N} \left( \boldsymbol{\mu}_n(\Theta), \sigma^2 \mathbf{I} \right) \quad (24)$$

$$\boldsymbol{\mu}_n(\Theta) = \mathbf{o}_r + \Delta\mathbf{o}_{i(n)} + \left( \mathbf{Z}_r + \Delta\mathbf{Z}_{i(n)} \right)\mathbf{b}_n + s_n \mathbf{1}_G \quad (25)$$

**Inference from paired UMI counts.** Consider the count matrices $\mathbf{M}_1 \in \mathbb{Z}^{G \times N}$ and $\mathbf{M}_2 \in \mathbb{Z}^{G \times N}$, with each pair of elements $m_{1,g,n}$ and $m_{2,g,n}$ corresponding to success and failure counts, respectively, in $m_{g,n} = m_{1,g,n} + m_{2,g,n}$ Bernoulli trials with success probability $p_{g,n}$. GEDI models $\mathbf{y}_n = (\text{logit}p_{1,n}, \ldots, \text{logit}p_{G,n})^\top \in \mathbb{R}^G$ as a latent variable:

$$m_{1,g,n} \sim \text{B} \left( m_{g,n}, \text{S} \left( y_{g,n} \right) \right) \quad (26)$$

$$\mathbf{y}_n | \boldsymbol{\mu}_n(\Theta) \sim \mathcal{N} \left( \boldsymbol{\mu}_n(\Theta), \sigma^2 \mathbf{I} \right) \quad (27)$$

$$\boldsymbol{\mu}_n(\Theta) = \mathbf{o}_r + \Delta\mathbf{o}_{i(n)} + \left( \mathbf{Z}_r + \Delta\mathbf{Z}_{i(n)} \right)\mathbf{b}_n + s_n \mathbf{1}_G \quad (28)$$

Here, S is the sigmoid function.

**Obtaining maximum a posteriori estimates of model parameters.** When the gene expression matrix $\mathbf{Y}$ is provided and no gene-level or sample-level prior information is available, GEDI uses a block coordinate descent algorithm to obtain maximum a posteriori estimates for

$\mathbf{Z}_r$, $\Delta\mathbf{Z}_i$, $\mathbf{o}_r$, $\Delta\mathbf{o}_i$ and $\mathbf{B}$. When $\mathbf{Y}$ is latent (i.e., when $\mathbf{M}$ or the pair $\mathbf{M}/\mathbf{M}'$ is provided), $\mathbf{Z}_r$ is latent (i.e., gene-level variables are provided), and/or $\Delta\mathbf{Z}_i$ and $\Delta\mathbf{o}_i$ are latent (i.e., sample-level variables are provided), GEDI uses expectation-maximization to obtain maximum a posteriori estimates (see Supplementary Methods for details).

## Datasets and preprocessing

All datasets used in the article were downloaded from the publication access codes, from the publication's online data repositories, or from public data repositories. Quality control and preprocessing steps were performed with the scuttle package[45] (v 1.0.4). For further details, see Supplementary Methods.

## Integration methods and benchmarks

We compared the integration performance of GEDI against several other methods, including Seurat[46], LIGER[40], Harmony[41], BBKNN[47], scVI[39], Scanorama[48], and CSS[38], as well as PCA (no integration) as a baseline. We ran each method following the documentation obtained from available tutorials or paper methods. Unless specified, we used the default parameters established by each package. For further details, see Supplementary Methods.

We measured the ability of each method to remove technical variability while preserving biological variation associated with the cell types. To do this, we followed the approach established by previous integration benchmark efforts[16,17], which used metrics that can be grouped into two broad categories: (a) removal of batch effects and (b) conservation of biological variance. Metrics from group (a) included alignment score (batch), iLISI, kBET, and ASW (batch), while group (b) metrics included alignment score (cell type), cLISI, ARI, NMI and ASW (cell type). For further details on each individual metric, see the Supplementary Methods section.

For each method, we defined an overall score that summarized the performance of the multiple metrics, following the approach established by Luecken et al.[16]. Briefly, we first rescaled the output of every metric to range from 0 to 1, which ensures that each metric is equally weighted within a partial score and has the same discriminative power. The rescaling was done using the transformation $y'=[y-\min(Y)]/[\max(Y)-\min(Y)]$. Then, we defined the 'Batch' score' and the 'Bio' score, representing the average for the metrics belonging to the groups (a) and (b) above, respectively. For a given integration method, we calculated the overall score as previously defined by Luecken et al.[16], where a weighted average of the two Bio and Batch scores was used, with a weight of 0.6 for the Bio score and 0.4 for the Batch score (integration metrics using different weights can be found in Supplementary Fig. 3). Additional details are found in Supplementary Methods.

## Differential expression analysis

We performed pseudobulk differential expression (DE) analysis on the COVID-19 data using DESeq2[49] (v.30.1). For each cell type and donor combination, we created a pseudobulk using the 'aggregateAcrossCells' function from scuttle[45]. Only cell types that were present in all three conditions (control, mild and severe COVID-19) were considered for DE analysis.

## Cluster-free differential expression benchmark

We conducted a systematic analysis to evaluate the performance of GEDI in clustering-free DGE analysis. Our assessment included the comparison of GEDI to two clustering-free DGE methods: LEMUR[24] and miloDE[25], as well as a comparison to DESeq2-based pseudobulk analysis. First, we developed a generative model that simulates cohort-level scRNA-seq data while preserving characteristics such as gene-gene and cell-cell correlations observed in real data, as discussed in detail in Supplementary Methods. Briefly, for each sample $i$, we simulate $K$ archetypes representing a set of vertices on the gene expression

manifold. If we represent the expression of each gene $g$ across the $K$ archetypes using the column vector $\boldsymbol{\gamma}_{g,i}\in\mathbb{R}^K$, then we have:

$$\boldsymbol{\gamma}_{g,i} \sim \mathcal{N}\left(\mathbf{X}_g\mathbf{h}_i + \mathbf{x}'_g\mathbf{h}_i\mathbf{1}_K, \boldsymbol{\Sigma}_g\right) \tag{29}$$

where $\mathbf{X}_g$ is a matrix of coefficients ($\mathbf{X}_g\in\mathbb{R}^{K\times L}$) that shows, for each of the $K$ archetypes, how the mean expression of gene $g$ is determined by each of the $L$ sample-level variables represented by the column vector $\mathbf{h}_i$. $\mathbf{x}'_g\in\mathbb{R}^L$ is a row vector that represents the global effects of sample-level variables on the observed abundance of gene $g$, e.g., through affecting ambient RNA abundances or other artifacts[50]. $\boldsymbol{\Sigma}_g\in\mathbb{R}^{K\times K}$ is a covariance matrix for gene $g$.

Each cell $n$ is then modeled as a probabilistic function of the weighted average of the archetypes of its sample:

$$y_{g,n} \sim \mathcal{N}\left(\mathbf{w}_n\boldsymbol{\gamma}_{g,i(n)}, \sigma_g^2\right) \tag{30}$$

where $\mathbf{w}_n\in\mathbb{R}_+^K$ is a row vector of weights connecting cell $n$ to each of the $K$ archetypes ($\Sigma_k w_{n,k}=1$), and $\sigma_g^2$ is a gene-specific variance. Thus, the ground-truth DE effect of the $L$ sample-level variables on gene $g$ in cell $n$ can be derived as the row-vector $\boldsymbol{\delta}_{n,g}=\mathbf{w}_n\mathbf{X}_g$.

Finally, for each cell $n$, the UMI counts are simulated as:

$$p_{g,n} = \frac{c^{y_{g,n}}}{\sum_{g=1}^{G} c^{y_{g,n}}} \tag{31}$$

$$\mathbf{m}_n \sim \text{Multinom}_G\left(M_n, \mathbf{p}_n\right) \tag{32}$$

where $c$ is the logarithmic basis for the simulated log-scale profiles and $M_n$ is the total UMI count for cell $n$. This framework allows us to simulate UMI counts starting from a given set of $\mathbf{X}_g$, $\boldsymbol{\Sigma}_g$, and $\sigma_g$ for all $g\in\{1,\ldots,G\}$, and $\mathbf{w}_n$ and $M_n$ for all $n\in\{1,\ldots,N\}$; we call these the simulation parameters, which can be modified to obtain different simulated datasets with varying numbers of cells, cell-cell or gene-gene correlation structures, cellular diversities, or ground-truth DE effects. In this work, we used the COVID-19 cohort 1 dataset as template to derive realistic simulation parameters, as described in Supplementary Methods, and generated a simulated dataset with the same number of cells and samples as those of the COVID-19 dataset. We then applied GEDI, LEMUR, miloDE, and DESeq2-based pseudobulk analysis to this dataset (see Supplementary Methods), and compared the differential expression estimates from each method to the ground truth DE vectors at the single-cell (GEDI and LEMUR), neighborhood (GEDI, LEMUR and miloDE) and cell-type levels (GEDI, LEMUR and DESeq2). For further details, see Supplementary Methods.

## Cell type signature analysis

For the generation of cell-type signatures in PBMC data, we performed DE analysis using DESeq2 for each donor. Our model compared the mean expression of a given cell type versus the average of other cell types, including the batch variable (different sequencing technologies) as a covariate in the model. We considered genes with FDR adjusted p-value < 0.05 and $\log_2$ fold-change >1 to define cell type markers. For a given donor, we defined a binary matrix that contained the cell-type markers, which was used as input for applying GEDI on the other donor.

To perform the enrichment of cell type signatures and TF activities, we applied the scoreMarkers function from scran[51] (v.1.30.0) using the inferred cell type/TF activities by GEDI.

## Gene regulatory network analysis

For analysis of transcription factor (TF) networks (in PBMCs), we downloaded the human DoRothEA gene-regulatory network[52] and restricted the TF-gene interactions to the high-confidence sets A, B,

and C, as previously defined. We then refined this TF-gene interaction matrix, using the whole blood data from GTEx[53], to obtain a generative gene regulatory model in which the expression profile of each gene is a function of the expression of the TFs that are linked to it. Specifically, the target gene expression matrix $\mathbf{Y} \in \mathbb{R}^{G \times N}$ (which is TMM-normalized[54] and converted to log-scale) is modeled as $\mathbf{Y} \sim \mathbf{CA}$, where $\mathbf{A} \in \mathbb{R}^{P \times N}$ is the matrix of the expression of $P$ transcription factors across $N$ whole blood samples from GTEx, and $\mathbf{C} \in \mathbb{R}^{G \times P}$ is a weighted regulatory network whose elements are restricted to have the same sign as the regulatory interactions obtained from DoRothEA ($c_{g,p} \geq 0$ for an activating interaction between transcription factor $p$ and gene $g$, $c_{g,p} \leq 0$ for an inhibitory interaction, and $c_{g,p} = 0$ for no interaction). $\mathbf{C}$ was further filtered to include only TFs (columns) that have at least 10 "substantial" regulatory interactions, where we define a substantial effect as having an absolute value > 0.1.

For analysis of post-transcriptional regulatory networks, we defined a regulatory network of RNA binding proteins (RBPs) and microRNAs (miRNAs) based on their potential interactions with mRNA 3′ UTRs. For the RBPs, we downloaded the set of known motifs for all human RBPs from CISBP-RNA[55], and filtered them to include only the non-redundant set of motifs described in ref. 56. Then, for each human gene, the transcript with the longest 3′ UTR was identified. Genes whose longest 3′ UTR was shorter than 650nt were removed, and the first 650nt of the 3′ UTR (i.e., the 650nt region immediately downstream of the stop codon) of the remaining genes was used for motif scanning, using AffiMx[57]. The miRNA regulons were obtained from ref. 34.

### Assessment of estimated TF activities

We evaluated the accuracy of GEDI at estimating changes in TF activity by analyzing a previously published single-cell perturbation dataset[28]. Our evaluation also included a comparison to decoupleR[58] (v.2.8.0), an ensemble of computational frameworks to infer biological activities. The methods we included in our evaluation were "aucell", "gsva", "mlm", "ora", "ulm", "viper", "wmean", and "wsum". For wmean and wsum, we utilized the normalized outputs "norm_wmean" and "norm_wsum".

The dataset consisted of two technical batches which were split and analyzed separately. One batch was used to generate a gene signature for each TF and also measure the degree of association of each TF with the principal axes of heterogeneity of the data, while the other was used to estimate TF activities using the learned gene regulatory effects. We opted to learn the gene signature of each TF from the data (as opposed to using an external gene regulatory network) to ensure that our results purely reflected the performance of the activity inference methods without the confounding effect of the quality of the external gene network. At the same time, by using one batch for learning gene-TF associations and the other for evaluating activity inferences, we aimed to avoid circularity. These steps were repeated by swapping the batches. To generate the gene signature of each TF, we performed differential expression analysis for each TF between the cells with perturbed TF expression versus unperturbed cells, using the scoreMarkers function from scran. The mean standardized Cohen's d log-fold-change was used to define the regulatory effect of a TF on each gene. To identify the TFs whose perturbation is associated with a significant change in the principal axes of heterogeneity of the single-cell RNA-seq data, we ran PCA on the normalized expression data and retrieved the first 40 principal components. Next, for each TF, we fitted a logistic regression model to predict the TF perturbation status of a cell using its the principal component scores, and used a likelihood-ratio test to compare this model to a reduced model with only an intercept.

Single-cell TF activities were estimated for each method using the learned gene-regulatory signatures. For methods that only accept a gene list for each TF (aucell, fgsea, gsva and ora), we defined the downstream targets of each TF as the set of genes with $\log_2$ fold-change < −0.3 in TF-perturbed cells in comparison to unperturbed cells. For the methods from decoupleR, normalized expression values were used as input, while for GEDI we used raw counts.

### Differential analysis of cell type-specific cassette exon splicing

To perform DE between GABAergic and Glutamatergic cells in the Tasic data, we applied limma[59] (v.3.46) using the imputed value of the latent splicing matrix from GEDI, which represents the logit of PSI (percent-spliced-in). We applied the lmFit and eBayes functions to obtain DE estimates using the default parameters.

GSEA analysis was performed using fgsea[60] (v.1.16), with the following arguments: minSize=15, eps=0, and maxSize=500. For the analysis of cell-type exon inclusion events using the Tasic data, we used the M5 ontology gene sets from MSigDB[61] (m5.all.v2022.1.Mm.symbols.gmt), using the t-statistics from limma as the gene ranks.

Sashimi plots were generated using sashimipy[62] (v 0.0.6). For the Tasic dataset, we selected a subset of 100 cells for each cell type and dataset combination, generated a merged BAM file, and provided it as input to sashimipy.

### Statistics and Reproducibility

Unless specified, cells were removed from the downloaded datasets based on standard quality control criteria (% of mitochondrial reads <20%; number of UMIs <1000, and number of detected genes <1000). No samples were collected in this study. No statistical method was used to predetermine sample size. Computational experiments and analysis are reproducible using the notebooks and code provided.

### Reporting summary

Further information on research design is available in the Nature Portfolio Reporting Summary linked to this article.

## Data availability

All datasets used in the article were downloaded from the publication access codes or from their online data repositories. PBMC data is available at GEO accession number GSE132044. Pancreas data is available at ArrayExpress accession number E-MTAB-5061 and at GEO accession number GSE84133. The Tabula Muris Bone Marrow data is available from the publication repository [https://figshare.com/articles/dataset/Processed_files_to_use_with_scanpy_/8273102]. The COVID-19 dataset is available at the European Genome-phenome Archive (EGA) under access number EGAS00001004571. Tasic dataset is available at GEO accession number GSE71585 and GEO accession number GSE115746. Faure dataset is available at GEO accession number GSE150150. LaManno dataset is available at the original author's repository [http://pklab.med.harvard.edu/velocyto/hgForebrainGlut/]. Genga dataset is available at Zenodo accession number 10.5281/zenodo.3564178 [https://zenodo.org/doi/10.5281/zenodo.3564178]. Preprocessed data, embeddings, and GEDI models can be accessed via Zenodo (DOIs: 10.5281/zenodo.8222039, 10.5281/zenodo.8222697, 10.5281/zenodo.11163741, and 10.5281/zenodo.11164776). A description of the Zenodo files is available in Supplementary Table 1. Source data are provided with this paper.

## Code availability

GEDI[63] is available via GitHub at https://github.com/csglab/GEDI and via Zenodo (DOI: 10.5281/zenodo.12761204). Reproducible notebooks for the analyses presented in this work can be found at https://github.com/csglab/GEDI_manuscript.

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

## Acknowledgements

We thank Adrien Osakwe for his help with an earlier iteration of COVID-19 pseudobulk analysis. This work was supported by the Canadian Institutes of Health Research (CIHR) grant PJT-173317, New Frontiers in Research Fund grant NFRFE-2019-00975, and resource allocations from Digital Research Alliance of Canada to HSN. AM is supported by a doctoral training award from Fonds de Recherche du Québec Santé. HSN holds a CIHR Canada Research Chair.

## Author contributions

Methodology and conceptualization: A.M. and H.S.N. Mathematical derivation: H.S.N, with contributions from T.L. Code implementation: A.M. and H.S.N. Analysis: A.M., with contributions from L.M.S. and H.S.N. Visualization: A.M. and H.S.N. Writing: A.M. and H.S.N. Review and editing: T.L. and L.M.S. Study supervision and direction: H.S.N.

## Competing interests

The authors declare no competing interests.
