## [Peer Review File · Nature Communications]

A unified model for interpretable latent embedding of multi-sample, multi-condition single-cell dataReviewer #1 (Remarks to the Author):

General comments:

First I would like to congratulate the authors on a very clean manuscript, package, and reproducible notebooks. Overall I found this paper to be inventive and useful to a wider audience. I think many bioinformatics/comp.bio people would be interested in trying this methodology out. To me, the biggest win for this method is its superior integration performance across datasets and latent factors. While I felt like the integration section was the strongest part of the manuscript, I believe that the other sections could be strengthened by the addition of more quantitative comparisons. I give more specific details below, but the ones that I think would be easiest to implement would be 1) instead of using Pearson correlations in the DGE analysis to instead calculate the sensitivity and specificity of GEDI to identify differentially expressed genes, this would better convince me that I should use GEDI over Seurat/Scanpy workflows that I am already familiar with. 2) For the TF analysis, provide the readers with which TFs you expect to be differential (from the literature or from the paper you take the COVID-19 data from), it's hard to judge whether or not GEDI is accurate or not with the results you have provided when I'm not sure which TFs should be differential, especially when it is not compared with the TFs identified in the source data publication. If possible, but not necessary, it might be useful to compare GEDI's performance when there are known knock-downs/outs of TFs.

Manuscript comments:

1. In the section "GEDI disentangles different sources of sample-to-sample variability" it is unclear how the effect of the technology is regressed out. Is this when you explicitly model it as a sample-level variable? Furthermore, if you take one of the two largest sources of variation and regressed one out, isn't it expected that the next largest source of variability would be from the donors? Furthermore, I don't understand how this is disentangling the sources of sample-to-sample variability if you are directly regressing out one source of variability. In this example, where sample ID and technical noise are orthogonal signals of variation, I can imagine this is easy to disentangle, but what about instances where two signals of variation happen to be correlated, would GEDI be able to disentangle them?
2. What is the justification for using the weighting of 0.6 for the Bio score and 0.4 for the Batch score? Does this choice change performance dramatically?
3. Using the COVID-19 dataset, how well would SVM perform using the gene expression of pseudobulks and training on one cohort and testing on the other cohort? Or taking the classical monocytes alone and performing the prediction task? Can you also include a shuffled or random baseline? Just to ensure that it isn't a sample-label bias issue that the performance is so high.
4. For figure 3e, it isn't fully clear, but I am assuming the mean vector field per cell type is the length of genes? Also, the maximum Pearson correlation is pretty low, with its maximum only being only 0.3 between the pseudobulk and GEDI. While the overall correlation may be low, I think what most people care about in DGE analysis is if the "top hits" are preserved. One could show this by seeing if in your transcriptomic vector field for a cell type, the genes with the largest change are the gene expected to be differentially expressed between COVID-19 and healthy. Also, is Pearson correlation appropriate in this analysis? I am not sure what the distribution of the vector is.
5. Figure 4a -- the adj. P-values are extremely significant, it may indicate that this test is not appropriate for this comparison. Would it be possible to also show the correlation value, not just the p-value?
6. Figure 4d -- it's very hard to understand the triangles within each square. Since I don't see a strong difference between the bottom and top triangle, would it be possible only to include one? Also having the size change makes it additionally difficult to interpret the triangles.
7. Figure 4d -- It's difficult to understand if you are finding what you expect. I understand that you find cell-type specific TF activity, but I'm unsure how accurate it is, especially with no other comparison. From the original paper, they state they find STAT3, CEBPD and CEBPE are predicted to be differential in the HLA-DR10 S100Ahi monocytes, but I don't find them depicted in your figure, is there a reason why they aren't depicted?
8. In Supplementary Figure 5, while I do see that the TF activity is cell-type specific, I don't believe I see the top TF lineage regulators for each cell type. I am not sure if this is because they

were't included in the signatures you used as prior knowledge or if they weren't significant, or if they weren't expected to be there in the first place. I think it would be helpful for the reader to include a table of your full results, as well as a quantification of the sensitivity/specificity of GEDI's ability to identify the expected cell-type specific TF regulators, if this was an expected finding. It may be easier to quantify the accuracy of GEDI estimating changes in TF activity using a Perturb-Seq dataset (like the one used in this paper "Robustness and applicability of transcription factor and pathway analysis tools on single-cell RNA-seq data")

9. In the section "Modeling the latent space of RNA splicing and stability with GEDI", I am not sure if 0.16 is very convincing that GEDI finds a significant signal. Is there a baseline you could use as a comparator? Maybe by shuffling the differential mRNA half-life measurements to show it's better than random guessing? I know that you provide a comparison with the bulk of 0.22, but I also don't have a sense of how significant this correlation is.

10. Why do you use limma and DESeq2 in different sections for DE analysis?

Notebook/Code comments:

1. Can the authors upload the Rmd files, not just the html, this makes them easier to run for others.
2. Can the authors provide an expected directory structure to the notebooks can be re-run easily by others.
3. I was able to download and install GEDI on my M1 mac pretty easily, good job!
4. It is not clear what this path is supposed to be "dir_data_hdf5<- paste0(dir_data, "pbmc_SCE/")" This is in all notebooks at the top, but I am unsure what it is supposed to point to since I don't see it in the zenodo links
5. It's helpful for people reading your code to know which methods come from your package, so namespacing them helps; for example writing "GEDI::plot_embedding"
6. You might want to consider adding your packages to Bioconductor, it will help it stay up-to-date and be easily installable even when R changes versions.
7. In the PBMC analysis, removing the effect of the donor, why are you only using the top 20 genes, and which "top" genes are they? This seems like much less than what people would use in common practice. I also don't see this mentioned in the manuscript; it may be useful to add.
8. In the COVID Notebook, what is the figure where the AUC is 0.87? How is this figure different than Figure 2e?

Thanks and congratulations on your work

Reviewer #2 (Remarks to the Author):

The authors presented GEDI, a method that performs integration of scRNA-seq data from multiple samples and conditions. In addition to performing integration, it can also detect differentially expressed genes across samples/conditions, calculating transcriptomics vector fields, and take two modalities as input where the ratio between the two modalities is of interest. The manuscript should be improved in terms of providing additional results to support its claims, highlighting its unique advantages and application scenarios, and providing information for readers to understand the results presented. The detailed comments are as following:

1. The manuscript claimed that one of the functions of GEDI is to perform denoising and/or imputation of single cell data at multiple locations, but no results on denoising/imputation are shown. Line 292 points to Figure 5d and Supp. Figure 9, but these figures do not show denoising/imputation results. In order to support this function of GEDI as part of this unified framework, results on denoising/imputation need to be provided, and results shall be compared to those of state-of-the-art methods. If the authors mean the denoising of PSI (as mentioned in ll 237-238) or imputing ratios between two measurements (ll 243-244, 252) it needs to be made clear and specific when mentioning denoising/imputation functions.

2. When comparing with existing methods on data integration performance, the results show comparison with a few methods compared in [Luecken, et al. 2021. "Benchmarking Atlas-Level

Data Integration in Single-Cell Genomics." Nature Methods]. The comparison should include top-performing methods from this benchmarking paper, such as scVI and Scanorama.

3. Regarding cluster-free DE analysis: the authors compared results on DE analysis with a pseudo-bulk approach. However, there are other cluster-free DE methods developed for single cell data, which can be more suitable baseline methods for GEDI. The results of cluster-free DE should be compared with these methods:

Ahlmann-Eltze, Constantin, and Wolfgang Huber. 2023. "Analysis of Multi-Condition Single-Cell Data with Latent Embedding Multivariate Regression." bioRxiv. doi:10.1101/2023.03.06.531268.

Missarova, Alsu, Emma Dann, Leah Rosen, Rahul Satija, and John Marioni. 2023. "Sensitive Cluster-Free Differential Expression Testing." bioRxiv. doi:10.1101/2023.03.08.531744.

4. In order to obtain reliable DE results, is there any requirement on the number of samples needed? What are the requirements on the distribution of samples across conditions?

5. Figure 4e-g shows a potential application of the TF activity calculated by GEDI. However, this part of the discussion (II 215-218) lacks evidence to show that the selected TFs that may be associated with the severe condition are biologically meaningful. To show the selected TFs are biologically meaningful, one can provide functional annotations of individual genes that are disease associated, or perform gene ontology analysis of a set of top-ranking TFs.

6. Regarding the TFs selected (discussed in II 215-218), can they be detected by simply performing differential expression analysis between severe and control conditions?

7. One major feature of GEDI is to calculate the transcriptomic vector fields. It would be helpful for users to determine whether this is a suitable tool to use if the authors can summarize the applications of the transcriptomic vector fields, one of which may be to detect differentially expressed genes across conditions. What are some other applications of this information that the users can take advantage of?

8. Some figures lack description on how they are made and which data was it made from, thus hinders understanding of these figures. (1) Figure 1a and Figure 1d, each sample corresponds to a point in these plots. Which data was used as input to the UMAP function to generate these plots? (2) Figure 2a: what is x-axis and y-axis? What are the background gray arrows and what are the longer arrows? (3) Figure 3b: what is the input to the UMAP function in this plot? The figure description says "the position of each cell represents the manifold embedding for the control condition", but it's not clear which version of the manifold embedding is used here. Is it a certain version of embedding learned in GEDI or simply a standard visualization of the original high-dimensional scRNA-seq data? (4) Figure 4c: we understand that the changes of gene expression levels can be calculated, but it's not clear how the changes for all genes in each single cell can be transformed into the arrow in the 2-D UMAP space.

9. Regarding the last part of the results (II 220-263), it would help if the major application scenarios of this type of analysis can be made more clear, that is, what are the outputs that users can obtain by using GEDI. From what I understand, GEDI can estimate PSI. GEDI also estimates the ratio between unspliced and spliced counts but the correlation between this ratio and mRNA half-life seems very low. The manuscript also shows that one can use both unspliced and spliced counts instead of the standard scRNA-seq data (spliced counts) to estimate regulator activities – in this regard, can the authors provide guidance or suggestions on when to use only scRNA-seq and when to use both spliced and unspliced counts?

Other comments:

Line 316: I would not use "GRN modeling" here as GRNs are used as prior knowledge to the model instead of part of the modeling process. Maybe "prior GRN information" is more appropriate.

Figure 2e description: "Top" and "Bottom" should be left and right.

The manuscript employs the terms "latent splicing space" (ll 231, 233, Supp Figure 9) which is widely used. It would be helpful to explain that it means and that is the biological interpretation of this latent space.

Code and Software Review:

1. Are you satisfied that all data and source code needed to reproduce the results of the paper have been made available?

Yes. The source code of the method is available on GitHub with the link <https://github.com/csglab/GEDI/>. The code to reproduce the results is also available on GitHub: https://github.com/csglab/GEDI_manuscript.

2. Are you satisfied that the results can be replicated using the code/software and dataset provided in the study?

Yes, the authors provide the code that generates the Figures and analysis results in the manuscript on GitHub (https://github.com/csglab/GEDI_manuscript).

3. Were you able to run the tool successfully?

Yes. I'm able to successfully install the tool using R, and run the sample tutorial on GitHub.

4. Was the code sufficiently documented to allow another researcher to follow the algorithm?

Yes. The GitHub site provides detailed documentation and tutorials on running the algorithm.

5. Can the software be run on a widely available operating system?

The method is written as an R package, and the authors provide the installation procedure of the package. I'm able to successfully install it on the Ubuntu system.

6. To your knowledge, do available tools or software exist that perform in a similar way to the reported software?

There is no tool that performs in a similar way to the reported software.

7. In cases when the source code is not provided but the mathematical description of the algorithm is; was the core mathematical algorithm sufficiently documented to allow another researcher to reproduce it?

The source code is provided and the mathematical description in the manuscript is clear enough.

Reviewer #3 (Remarks to the Author):

I co-reviewed this manuscript with one of the reviewers who provided the listed reports as part of the Nature Communications initiative to facilitate training in peer review and appropriate recognition for co-reviewers.

Please find below our point-by-point responses to reviewers' comments. Our responses are in blue. In addition, in the manuscript and the associated supplementary materials, we have highlighted the major sections of the text that have been modified compared to the previous version.

Here is a summary of the changes:

1. Key new analyses:
 - a. We have included new analyses in response to reviewers #1 and #2, regarding the comparison of performance of cluster-free differential expression methods. These results include the development of a generative model to simulate cohort-level scRNA-seq data as well as the comparison of GEDI differential expression estimates to other methods.
 - b. We have incorporated new analyses in response to reviewer #1, where we perform a quantitative assessment of the TF activities estimated by GEDI and other methods using a single cell perturbation dataset.
 - c. We have included new analyses in response to reviewer #2, incorporating additional methods in the integration benchmark.
 - d. We have added new analyses in response to reviewer #2 to evaluate the performance of GEDI in imputation of the ratios between two measurements.
2. We have made additional edits to the text, figures, and figure legends to address the reviewers' concerns about clarity and presentation of the results.
3. We have edited and expanded the reproducible notebooks of the manuscript in GitHub to improve the clarity and reproducibility of the analyses.

Reviewer #1 (Remarks to the Author):

General comments:

First I would like to congratulate the authors on a very clean manuscript, package, and reproducible notebooks. Overall I found this paper to be inventive and useful to a wider audience. I think many bioinformatics/comp.bio people would be interested in trying this methodology out. To me, the biggest win for this method is its superior integration performance across datasets and latent factors. While I felt like the integration section was the strongest part of the manuscript, I believe that the other sections could be strengthened by the addition of more quantitative comparisons. I give more specific details below, but the ones that I think would be easiest to implement would be 1) instead of using pearson correlations in the DGE analysis to instead calculate the sensitivity and specificity of GEDI to identify differentially expressed genes, this would better convince me that I should use GEDI over Seurat/Scanpy workflows that I am already familiar with. 2) For the TF analysis, provide the readers with which TFs you expect to be differential (from the literature or from the paper you take the COVID-19 data from), it's hard to judge whether or not GEDI is accurate or not with the results you have provided when I'm not sure which TFs should be differential, especially when it is not compared with the TFs identified in the source data publication. If possible, but not necessary, it might be useful to compare GEDI's performance when there are known knock-downs/outs of TFs.

We thank the reviewer for the positive assessment of the manuscript and the view that GEDI will be useful to other members of the bioinformatics and computational biology community. As outlined below, we have performed additional analyses and edited the text to address the reviewer's concerns.

Manuscript comments:

1. In the section “GEDI disentangles different sources of sample-to-sample variability” it is unclear how the effect of the technology is regressed out. Is this when you explicitly model it as a sample-level variable? Furthermore, if you take one of the two largest sources of variation and regressed one out, isn't it expected that the next largest source of variability would be from the donors? Furthermore, I don't understand how this is disentangling the sources of sample-to-sample variability if you are directly regressing out one source of variability. In this example, where sample ID and technical noise are orthogonal signals of variation, I can imagine this is easy to disentangle, but what about instances where two signals of variation happen to be correlated, would GEDI be able to disentangle them?

We have now revised the text to clarify the points raised by the reviewer. Briefly, during this step, we are not providing sample-level variable information to GEDI. Our objective is to show that the sample-specific distortion parameters learned by GEDI capture variability that is introduced by a combination of different sample-level characteristics, in this case, the donor and technology effects. Further details of how the effect of technology or donor is regressed out can be found in the legend for Figure 2 and, in more details, in the legend for Supplementary Figure 1.

We have now edited the title of the section to “*GEDI captures different sources of sample-to-sample variability*”, avoiding the term “disentangle”, which was confusing.

Also, we have edited the text to explain that no sample-level variable information was given during this step.

Page 4: “*To assess the ability of GEDI to capture sample-to-sample variability, we applied it to a dataset of peripheral blood mononuclear cells (PBMCs) from two donors¹⁵ profiled using different scRNA-seq technologies. We applied GEDI without including sample-level variable information, so the model was oblivious to the biological and technical characteristics of the samples.*”

We have also edited the legend of Figure 2 to indicate that the effect of technology was regressed out from sample-specific manifold parameters post-hoc.

These analyses, together, show that the sample-specific distortions learned by GEDI are functions of a combination of technical and biological variables; we hope the revised text better reflects this point.

2. What is the justification for using the weighting of 0.6 for the Bio score and 0.4 for the Batch score? Does this choice change performance dramatically?

We have now clarified in the Methods section (page 16) that we simply gave a weight of 0.6 following the formula for the overall integration score defined previously by Luecken et al (2018). Per Luecken et al., the basis for establishing a higher weight to the biological score was to set a greater emphasis on the preservation of the variability that corresponds to cell type differences. To evaluate the effect of the choice of weights, we have now included a re-assessment of the overall integration score in the new Supplementary Figure 3. We observe that the performance of GEDI as one of the top performing methods remains stable irrespective of the choice of weights in the PBMC and Pancreas datasets, and while we observe a higher variability in the Tabula Muris dataset, this occurs for all methods analyzed.

3. Using the COVID-19 dataset, how well would SVM perform using the gene expression of pseudobulks and training on one cohort and testing on the other cohort? Or taking the classical monocytes alone and performing the prediction task? Can you also include a shuffled or random baseline? Just to ensure that it isn't a sample-label bias issue that the performance is so high.

We have now included a comparison of the performance of SVM using different predictors as suggested by the reviewer (please see the new Supplementary Figure 2). This analysis contrasts the performance of using sample-specific transformations from GEDI versus employing the pseudo-bulk normalized data from DESeq2, as well as an evaluation of the impact of the number of most variable features used in performance.

We observe that utilizing the sample-specific transformations from GEDI offers a better performance (0.97 mean AUROC across the 2 cohorts and different number of most variable features) compared to training using a pseudobulk of all cell types (0.48 mean AUROC) or when restricting the pseudobulk to a specific cell type (0.75 mean AUROC). A random baseline using shuffled labels in the training set, as suggested by the reviewer, shows that the mean AUROCs approximate a random guessing for all methods (also shown in Supplementary Figure 2). These results are now cited on page 5.

4. For figure 3e, it isn't fully clear, but I am assuming the mean vector field per cell type is the length of genes?

Yes. We have edited the figure legend to clarify this point (page 27):

"The heatmap shows the Pearson correlation values between the GEDI-based mean vectors (columns) and the pseudobulk-based DE vectors (rows) between cell type pairs, for the comparison of mild COVID-19 vs. control cases in cohort 1 (each element of the heatmap represents Pearson correlation across all genes)"

Also, please see our new Supplementary Figure 4 and our response to comment 2.7, where we broaden our explanation of the transcriptomic vector field.

Also, the maximum Pearson correlation is pretty low, with its maximum only being only 0.3 between the pseudobulk and GEDI. While the overall correlation may be low, I think what most people care about in DGE analysis is if the "top hits" are preserved. One could show this by seeing if in your transcriptomic vector field for a cell type, the genes with the largest change are the gene expected to be differentially expressed between COVID-19 and healthy. Also, is Pearson correlation appropriate in this analysis? I am not sure what the distribution of the vector is.

We thank the reviewer for this suggestion. We have now performed a systematic benchmarking of the ability of GEDI in clustering-free differential expression analysis using simulated data, in order to enable comparison to a known ground truth. Following the reviewer's suggestion, we have included metrics showing the performance of GEDI in preserving the "top hits" (genes with the highest degree of differential expression). As shown in the new Figure 4, GEDI (hyperplane) is able to recover the top up-regulated and down-regulated genes in each cell with median AUROC of ~0.78, considerably higher than recent competing methods. We have provided more information about this benchmarking analysis in our response to comment 2.3, where we specifically discuss the comparison of different methods for performing clustering-free DE analysis.

5. Figure 4a -- the adj. P-values are extremely significant, it may indicate that this test is not appropriate for this comparison. Would it be possible to also show the correlation value, not just the p-value?

We agree with the reviewer. We have now updated Figure 5a (previously Figure 4a), Supplementary Figure 8, Supplementary Figure 9 and Supplementary Figure 17 showing the mean AUROC scores instead of P-values. We chose the AUROC as it has been introduced previously by Amezcua et al (2020, doi: 10.1038/s41592-019-0700-8) as a measure of the specificity of markers in single-cell analysis.

6. Figure 4d -- it's very hard to understand the triangles within each square. Since I don't see a strong difference between the bottom and top triangle, would it be possible only to include one? Also having the size change makes it additionally difficult to interpret the triangles.

We agree with the reviewer and have updated Figure 5d (previously Figure 4d) to include only the cosine similarity. However, we have kept the size scale, as we believe that knowing the cell types in which a given TF has higher expression provides relevant information to prioritize candidate TFs.

7. Figure 4d -- It's difficult to understand if you are finding what you expect. I understand that you find cell-type specific TF activity, but I'm unsure how accurate it is, especially with no other comparison. From the original paper, they state they find STAT3, CEBPD and CEBPE are predicted to be differential in the HLA-DRI S100Ahi monocytes, but I don't find them depicted in your figure, is there a reason why they aren't depicted?

CEBPE was not part of the confident interactions from the DoRothEA (v.1.2.2) gene-regulatory network, while CEBPD was removed during the refinement of interactions of the TF-gene matrix. STAT3 was included in the analysis but is not shown in Figure 4d due to low correlation (< 0.25) between the inferred activity and mRNA abundance. Therefore, these TFs happen to be excluded mostly due to uncertainties in the source gene-regulatory network. We have, however, included additional analyses to establish potential biological relevance of other TFs that we found in our analyses, as shown in the new Supplementary Figure 13 and discussed on page 8. Specifically, we now highlight the strong up-regulation of *SPI1* mRNA in monocytes in severe COVID-19 compared to healthy controls, consistent with our activity gradient analysis, as well as enrichment of *SPI1*, *CEBPA* and *SP1* targets in immune-related pathways. Please also see our response to the next comment regarding benchmarking the ability of GEDI in inferring TF activities.

8. In Supplementary Figure 5, while I do see that the TF activity is cell-type specific, I don't believe I see the top TF lineage regulators for each cell type. I am not sure if this is because they weren't included in the signatures you used as prior knowledge or if they weren't significant, or if they weren't expected to be there in the first place. I think it would be helpful for the reader to include a table of your full results, as well as a quantification of the sensitivity/specificity of GEDI's ability to identify the expected cell-type specific TF regulators, if this was an expected finding. It may be easier to quantify the accuracy of GEDI estimating changes in TF activity using a Perturb-Seq dataset (like the one used in this paper "Robustness and applicability of transcription factor and pathway analysis tools on single-cell RNA-seq data")

We have now provided an additional Supplementary Data Table 2, which includes the mean AUROC values per cell type for all TFs that were analyzed.

To perform a quantitative assessment of the accuracy of GEDI at estimating changes in TF activity, we have now included a new Supplementary Figure 11, where we benchmark TF activity using a single-cell perturbation dataset as suggested by the reviewer. Our results show that, for the TFs that drive a significant cell state shift along the principal axes of heterogeneity, GEDI is one of the top 2 methods in terms of correct inference of TF activity. As we discuss in the manuscript (page 7), this observation is consistent with GEDI's model, in which the principal axes of variation are modeled as a function of the gene-regulatory network, and therefore TFs whose variable activity drives these axes are expected to be captured by GEDI.

9. In the section "Modeling the latent space of RNA splicing and stability with GEDI", I am not sure if 0.16 is very convincing that GEDI finds a significant signal. Is there a baseline you could use as a comparator? Maybe by shuffling the differential mRNA half-life measurements to show it's better than random guessing? I know that you provide a comparison with the bulk of 0.22, but I also don't have a sense of how significant this correlation is.

We thank the reviewer for the suggestion of establishing a baseline. We have added a panel to Supplementary Figure 17, where we generated a null distribution of the Pearson correlation estimates by shuffling the pseudo-time labels when estimating differential stability using GEDI. The mean Pearson correlation of the null distribution is 0.002, in contrast to 0.16 obtained with GEDI using unshuffled scRNA-seq data.

10. Why do you use limma and DESeq2 in different sections for DE analysis?

For pseudo-bulk differential expression analysis, given that the observed values to be modeled were count data, we used DESeq2. However, DESeq2 could not be utilized to detect cell type-specific splicing events because the GEDI-inferred logit-PSI values were not count data. Instead, we chose limma, which can be fitted to log-normalized values.

Notebook/Code comments:

1. Can the authors upload the Rmd files, not just the html, this makes them easier to run for others.

We have now included the .Rmd files in the GitHub repository.

2. Can the authors provide an expected directory structure to the notebooks can be re-run easily by others.

We have now provided a description of the directories and data needed to run each step of the notebooks in the GitHub pages.

3. I was able to download and install GEDI on my M1 mac pretty easily, good job!

We thank the reviewer for the positive assessment.

4. It is not clear what this path is supposed to be "`dir_data_hdf5<- paste0(dir_data, "pbmc_SCE/")`" This is in all notebooks at the top, but I am unsure what it is supposed to point to since I don't see it in the zenodo links

The path establishes the input dataset used in each notebook. To clarify this, we have now included an additional notebook specifying the pre-processing steps that were made to each dataset, including the download and quality control steps.

5. It's helpful for people reading your code to know which methods come from your package, so namespacing them helps; for example writing "GEDI::plot_embedding"

We have now edited this in the notebooks.

6. You might want to consider adding your packages to Bioconductor, it will help it stay up-to-date and be easily installable even when R changes versions.

We appreciate the reviewer suggestion. We plan to submit GEDI to Bioconductor soon.

7. In the PBMC analysis, removing the effect of the donor, why are you only using the top 20 genes, and which "top" genes are they? This seems like much less than what people would use in common practice. I also don't see this mentioned in the manuscript; it may be useful to add.

We apologize for the omission of the number of most variable features used in Figure 2a; we have now updated the figure legend to clarify this point (page 26). To examine the effect of increasing the number of most variable features, we have now added a new Supplementary Figure 2. We observe that the clustering patterns of the sample-specific transformations remains consistent with our previous observation, irrespective of the number of most variable features used.

8. In the COVID Notebook, what is the figure where the AUC is 0.87? How is this figure different than Figure 2e?

The AUROC of 0.87 corresponds to the training score on cohort2, while Figure 2e shows the test score of cohort 1 (trained on cohort2) and cohort 2 (trained on cohort1).

Thanks and congratulations on your work

Reviewer #2 (Remarks to the Author):

The authors presented GEDI, a method that performs integration of scRNA-seq data from multiple samples and conditions. In addition to performing integration, it can also detect differentially expressed genes across samples/conditions, calculating transcriptomics vector fields, and take two modalities as input where the ratio between the two modalities is of interest. The manuscript should be improved in terms of providing additional results to support its claims, highlighting its unique advantages and application scenarios, and providing information for readers to understand the results presented. The detailed comments are as following:

We thank the reviewer for their comments and suggestions. We have now performed additional analyses and edited the manuscript to address the concerns of the reviewer, as outlined below.

1. The manuscript claimed that one of the functions of GEDI is to perform denoising and/or imputation of single cell data at multiple locations, but no results on denoising/imputation are shown. Line 292 points to Figure 5d and Supp. Figure 9, but these figures do not show denoising/imputation results. In order to support this function of GEDI as part of this unified framework, results on denoising/imputation need to be provided, and results shall be compared to those of state-of-the-art methods. If the authors mean the denoising of PSI (as mentioned in ll 237-238) or imputing ratios between two measurements (ll 243-244, 252) it needs to be made

clear and specific when mentioning denoising/imputation functions.

We thank the reviewer for pointing out this ambiguity. We have now made two main changes to address this issue:

First, we have compared the performance of GEDI vs. that of two other imputation methods, SAVER and MAGIC, for imputing the ratios between two measurements, using simulated data. However, since SAVER and MAGIC are not really designed to impute ratios, we did not present these results as “benchmarking” in the manuscript; instead, on pages 8-9 we write:

“[...] simulations [show] that GEDI can impute ground truth ratios from paired, sparse counts, while a naïve estimator provides ratios that are almost completely uncorrelated with the ground truth (Supplementary Figure 16a-e). Using the naïve estimator as input for two existing single-cell imputation methods^{32, 33} slightly improved its correlation with the ground truth, but GEDI substantially outperformed them in recovering the ground truth (Supplementary Figure 16f-g).”

Secondly, as suggested by the reviewer, we now make it explicitly clear that our work does not include the analyses necessary to show superior performance of GEDI in gene-level expression imputation/denoising compared to existing methods:

Page 10: *“As shown in Supplementary Figure 16, our simulation results underline the unique ability of GEDI to impute the ratios of paired observations (such as spliced vs. unspliced mRNA abundances), but it remains to be tested whether GEDI’s imputations for gene-level expression values are also competitive with existing single-cell imputation methods.”*

2. When comparing with existing methods on data integration performance, the results show comparison with a few methods compared in [Luecken, et al. 2021. “Benchmarking Atlas-Level Data Integration in Single-Cell Genomics.” Nature Methods]. The comparison should include top-performing methods from this benchmarking paper, such as scVI and Scanorama.

We have now included Scanorama and scVI in the integration benchmark. As shown in our updated Figure 2c, we observe that GEDI remains as one of the top performing methods in most benchmarking datasets.

3. Regarding cluster-free DE analysis: the authors compared results on DE analysis with a pseudo-bulk approach. However, there are other cluster-free DE methods developed for single cell data, which can be more suitable baseline methods for GEDI. The results of cluster-free DE should be compared with these methods:

Ahlmann-Eltze, Constantin, and Wolfgang Huber. 2023. “Analysis of Multi-Condition Single-Cell Data with Latent Embedding Multivariate Regression.” bioRxiv. doi:10.1101/2023.03.06.531268.

Missarova, Alsu, Emma Dann, Leah Rosen, Rahul Satija, and John Marioni. 2023. “Sensitive Cluster-Free Differential Expression Testing.” bioRxiv. doi:10.1101/2023.03.08.531744.

We have now performed a systematic analysis to compare the performance of GEDI to that of LEMUR and miloDE. Specifically, we have now included a new Figure 4, in which we use a simulated cohort-level scRNA-seq data as ground truth to compare GEDI’s differential expression estimates against other cluster-free DE methods. First, as shown in Figure 4a and described in more details in Supplementary Methods, we developed a generative model that simulates cohort-level scRNA-seq data while preserving characteristics such as gene-gene and cell-cell correlations observed in real data, providing us with a ground truth for condition-associated

differential expression estimates per individual cell. Then, we compared the DE estimates of GEDI to this ground truth, along with two other cluster-free DE methods (LEMUR and miloDE), as well as a pseudobulk method (DESeq2). Our new results, presented in Figure 4b-d, show that GEDI outperforms the other methods at the single-cell, neighborhood and cell type levels, using two different metrics (Pearson correlation, and AUROC values for distinguishing up- or down-regulated genes from invariable genes).

4. In order to obtain reliable DE results, is there any requirement on the number of samples needed? What are the requirements on the distribution of samples across conditions?

We have not performed a systematic analysis to establish such requirements. However, we recognize that the power and reliability to detect DE estimates at the single-cell level are most likely dataset-specific, as multiple factors are involved. We have now added a brief section in Discussion to underscore the need to consider these factors when performing DE analysis:

Page 10: *"We note, however, that more extensive analyses are needed to better understand the effects of factors that may influence the performance of GEDI and other DE analysis methods, including the number of differentially expressed genes per cell, the magnitude of their differential expression, cell-cell and gene-gene correlation structures, inter- and intra-sample variances, the number of samples, the number of cells per sample, and the sequencing depth."*

5. Figure 4e-g shows a potential application of the TF activity calculated by GEDI. However, this part of the discussion (ll 215-218) lacks evidence to show that the selected TFs that may be associated with the severe condition are biologically meaningful. To show the selected TFs are biologically meaningful, one can provide functional annotations of individual genes that are disease associated, or perform gene ontology analysis of a set of top-ranking TFs.

As suggested by the reviewer, we have now added a new Supplementary Figure 13, where we perform enrichment analysis of the top targets of SPI1, SP1 and CEBPA. This analysis shows enrichment of pathways involved in immune related processes, including innate immune system response, neutrophil degranulation and antigen processing-cross presentation. In addition, we have also performed a more systematic benchmarking of GEDI's TF activity inferences, as discussed in our response to comment 1.8 and showed in the new Supplementary Figure 11.

6. Regarding the TFs selected (discussed in ll 215-218), can they be detected by simply performing differential expression analysis between severe and control conditions?

As shown in our new Supplementary Figure 13, only SPI1 is upregulated in the severe COVID-19 versus control in the relevant cell types (different subsets of monocytes and neutrophils), by both DESeq2 and GEDI analysis.

7. One major feature of GEDI is to calculate the transcriptomic vector fields. It would be helpful for users to determine whether this is a suitable tool to use if the authors can summarize the applications of the transcriptomic vector fields, one of which may be to detect differentially expressed genes across conditions. What are some other applications of this information that the users can take advantage of?

We thank the reviewer for this suggestion. We have now added a brief text to the Discussion section to mention some of the applications of transcriptomic vector fields, while noting that these are potential applications that we have not yet tested with GEDI.

Page 11: “[...] the transcriptomic vector fields obtained by GEDI may have applications beyond clustering-free DE analysis. For example, earlier studies⁴³ have shown the utility of transcriptomic vector fields in prediction of cell fate transitions if the vector field represents “velocity” (gene expression change as a function of time). Furthermore, as GEDI’s vector field extends beyond the regions of the manifold that is occupied by observed cells, it may provide an opportunity for counterfactual prediction in previously unobserved cell types, such as prediction of response to specific perturbations^{44, 45}. These potential abilities, however, remain currently untested.”

8. Some figures lack description on how they are made and which data was it made from, thus hinders understanding of these figures.

(1) Figure 1a and Figure 1d, each sample corresponds to a point in these plots. Which data was used as input to the UMAP function to generate these plots?

To generate Figures 2a and 2b, we used the sample-specific distortion matrices that GEDI learns as input to PCA and UMAP, as detailed in the Figure legend. We also refer the reader to Supplementary Figure 1, which has a more extended description of the analysis. A complete description of how to access the sample-specific distortion matrices from the GEDI model can be found in the reproducible notebooks: https://csglab.github.io/GEDI_manuscript/notebooks/pbmc_analysis.html ; https://csglab.github.io/GEDI_manuscript/notebooks/COVID19_bothCohorts.html.

(2) Figure 2a: what is x-axis and y-axis? What are the background gray arrows and what are the longer arrows?

We have now edited the Figure legend to provide a clearer explanation of the figure. For Figure 1a, the axes represent gene-expression measurements (e.g. expression of gene 1 vs expression of gene 2). The background gray arrows in Figure 1a symbolize the mapping between cell state and observed gene expression profiles, which is provided by the sample-specific decoder functions Ψ_1 and Ψ_2 .

(3) Figure 3b: what is the input to the UMAP function in this plot? The figure description says “the position of each cell represents the manifold embedding for the control condition”, but it’s not clear which version of the manifold embedding is used here. Is it a certain version of embedding learned in GEDI or simply a standard visualization of the original high-dimensional scRNA-seq data?

We thank the reviewer for pointing out this ambiguity. The figure legend now more clearly specifies what embedding is used here:

“[...] the input for the UMAP was the low-dimensional projection of each cell on the manifold of the healthy group, as learned by GEDI (i.e., the reference manifold plus distortions associated with the control condition)”

(4) Figure 4c: we understand that the changes of gene expression levels can be calculated, but it’s not clear how the changes for all genes in each single cell can be transformed into the arrow in the 2-D UMAP space.

We apologize for our omission to provide the necessary details of our vector field visualization; we have now added a description in the revised Supplementary Methods. Briefly, to obtain the arrows in the 2D UMAP, for each single cell, we provide two “copies” as input to the UMAP algorithm: the projection of the cell on the manifold of the control group (as indicated in our response to the above comment), and its projection on the manifold of the target group (e.g., COVID-19 group). This is analogous to providing the UMAP algorithm with the counterfactual expression profile of each cell before and after COVID-19. Both copies are then projected onto the 2D space by UMAP; what we visualize in our plots is an arrow that connects the 2D projection of each control cell to its counterfactual COVID-19 cell. For a less cluttered visualization, we also group the cells that are near each other in the UMAP embedding, and plot the average of their associated vectors (averaged after UMAP projection). This method follows the suggestion by La Manno et al. (<https://doi.org/10.1038/s41586-018-0414-6>) for visualizing RNA velocity (which is also a vector field): “A variety of techniques can be used to visualize the velocity estimates in low dimensions. The observed and extrapolated cell states can be jointly embedded in a common low-dimensional space [...]”.

In addition to the details provided in the Supplementary Methods, we have also added a brief description in the figure legend (page 27):

“Arrows are obtained by jointly embedding, in the UMAP space, the (extrapolated) gene expression profiles of each cell in the control and severe COVID-19 condition (see Supplementary Methods for details).”

9. Regarding the last part of the results (II 220-263), it would help if the major application scenarios of this type of analysis can be made more clear, that is, what are the outputs that users can obtain by using GEDI. From what I understand, GEDI can estimate PSI. GEDI also estimates the ratio between unspliced and spliced counts but the correlation between this ratio and mRNA half-life seems very low. The manuscript also shows that one can use both unspliced and spliced counts instead of the standard scRNA-seq data (spliced counts) to estimate regulator activities – in this regard, can the authors provide guidance or suggestions on when to use only scRNA-seq and when to use both spliced and unspliced counts?

As pointed out by the reviewer, when provided with two quantities, one output of GEDI will be the imputed ratio (such as PSI) for each event in each cell. As schematically shown in Supplementary Figure 14, the interpretation of this ratio depends on the type of data to which the model is fitted, including exon inclusion rates, RNA stability, translation efficiency, ChIP enrichment and 5mC methylation ratio. Furthermore, GEDI provides a batch-corrected low-dimensional embedding of each cell, which can be used to identify cells for which the (latent) ratios are similar. The variability in the interpretation of the imputed ratio also extends to the interpretation (and usage) of network activity inferences: for example, if spliced/unspliced counts are provided, the imputed ratio will represent an estimate of mRNA stability, which can be used to infer the activity of regulatory factors that modulate mRNA stability (such as RBPs and miRNAs). These points are now briefly discussed in the Discussion section (page 10), as summarized below:

“We expect this functionality to be useful for analysis of other single-cell modalities that represent the ratio of two biological measurements, as summarized in Supplementary Figure 14, enabling a range of analyses based on those modalities, including batch correction and cluster/cell type analysis (similar to Figure 6a-c).”

“As shown in Supplementary Figure 16, our simulation results underline the unique ability of GEDI to impute the ratios of paired observations (such as spliced vs. unspliced mRNA abundances) [...]”

“This flexible framework enables the study of different types of regulatory mechanisms depending on the observations modelled. For example, it can be employed for the study of transcriptional regulators if gene-level counts are given as input (e.g., Supplementary Figure 9), or for the analysis of regulatory networks that modulate mRNA stability if paired spliced and unspliced transcript counts are provided (e.g., Supplementary Figure 17c).”

Other comments:

Line 316: I would not use “GRN modeling” here as GRNs are used as prior knowledge to the model instead of part of the modeling process. Maybe “prior GRN information” is more appropriate.

We have now edited this line in the manuscript to reflect that the GRNs are used as prior knowledge to the model: *“Overall, the framework presented here unifies a range of concepts that are central to single-cell data analysis, including [...] pathway and GRN activity analysis using prior information [...]”*

Figure 2e description: “Top” and “Bottom” should be left and right.

We apologize for the error; we have now fixed this.

The manuscript employs the terms “latent splicing space” (ll 231, 233, Supp Figure 9) which is widely used. It would be helpful to explain that it means and that is the biological interpretation of this latent space.

We have now edited the text to clarify the meaning of the term “latent splicing space”:

“[...] latent splicing space learned by GEDI, which represents the lower dimensional projection of the cells based on their (unobserved) cassette exon PSI values [...]”

Code and Software Review:

1. Are you satisfied that all data and source code needed to reproduce the results of the paper have been made available?

Yes. The source code of the method is available on GitHub with the link <https://github.com/csglab/GEDI/>. The code to reproduce the results is also available on GitHub: https://github.com/csglab/GEDI_manuscript.

2. Are you satisfied that the results can be replicated using the code/software and dataset provided in the study?

Yes, the authors provide the code that generates the Figures and analysis results in the manuscript on GitHub (https://github.com/csglab/GEDI_manuscript).

3. Were you able to run the tool successfully?

Yes. I'm able to successfully install the tool using R, and run the sample tutorial on GitHub.

4. Was the code sufficiently documented to allow another researcher to follow the algorithm?

Yes. The GitHub site provides detailed documentation and tutorials on running the algorithm.

5. Can the software be run on a widely available operating system?

The method is written as an R package, and the authors provide the installation procedure of the package. I'm able to successfully install it on the Ubuntu system.

6. To your knowledge, do available tools or software exist that perform in a similar way to the reported software?

There is no tool that performs in a similar way to the reported software.

7. In cases when the source code is not provided but the mathematical description of the algorithm is; was the core mathematical algorithm sufficiently documented to allow another researcher to reproduce it?

The source code is provided and the mathematical description in the manuscript is clear enough.

Reviewer #3 (Remarks to the Author):

I co-reviewed this manuscript with one of the reviewers who provided the listed reports as part of the Nature Communications initiative to facilitate training in peer review and appropriate recognition for co-reviewers.

Reviewer #1 (Remarks to the Author):

I thank the authors for their detailed responses to each of my comments. I believe that the authors have fully addressed all issues I raised previously.

My only comment is regarding my third comment:

Original comment from me: 3. Using the COVID-19 dataset, how well would SVM perform using the gene expression of pseudobulks and training on one cohort and testing on the other cohort? Or taking the classical monocytes alone and performing the prediction task? Can you also include a shuffled or random baseline? Just to ensure that it isn't a sample-label bias issue that the performance is so high.

Authors response: We have now included a comparison of the performance of SVM using different predictors as suggested by the reviewer (please see the new Supplementary Figure 2). This analysis contrasts the performance of using sample-specific transformations from GEDI versus employing the pseudo-bulk normalized data from DESeq2, as well as an evaluation of the impact of the number of most variable features used in performance.

We observe that utilizing the sample-specific transformations from GEDI offers a better performance (0.97 mean AUROC across the 2 cohorts and different number of most variable features) compared to training using a pseudobulk of all cell types (0.48 mean AUROC) or when restricting the pseudobulk to a specific cell type (0.75 mean AUROC). A random baseline using shuffled labels in the training set, as suggested by the reviewer, shows that the mean AUROCs approximate a random guessing for all methods (also shown in Supplementary Figure 2). These results are now cited on page 5.

My response to the authors: I agree with what is stated in the paper, "pseudobulk-based features did not generalize well across cohorts". But I think that stating "pseudobulk of all cell types (0.48 mean AUROC) or when restricting the pseudobulk to a specific cell type (0.75 mean AUROC)." Misses the fact that some of the pseudobulks work very competitively with GEDI, but it doesn't generalize well. Which I might guess that it is due to differences in cell type proportions. But no further changes are needed, I agree with what is stated in the paper.

Reviewer #1 (Remarks on code availability):

All previous comments regarding the code have been addressed by the authors.

Reviewer #2 (Remarks to the Author):

My concerns have been addressed.

Reviewer #3 (Remarks to the Author):
